



# A wind-driven snow redistribution module for Alpine3D v3.3.0: Adaptations designed for downscaling ice sheet surface mass balance

Eric Keenan[1], Nander Wever[1], Jan T. M. Lenaerts[1], and Brooke Medley[2]

[1]Department of Atmospheric and Oceanic Sciences, University of Colorado, Boulder, CO, USA
[2]Cryospheric Sciences Laboratory, NASA Goddard Space Flight Center, Greenbelt, MD, USA

**Correspondence:** Eric Keenan (eric.keenan@colorado.edu)

**Abstract.** Ice sheets gain mass via snow accumulation at the ice sheet surface, which is the primary component of surface mass balance. On the Antarctic ice sheet, winds redistribute snow resulting in surface mass balance that is variable in both space and time. Representing wind-driven snow redistribution processes in models is critical for local assessments of surface mass balance, repeat altimetry studies, and interpretation of ice core accumulation records. To this end, we have adapted Alpine3D, an existing distributed snow modeling framework, to downscale Antarctic surface mass balance to horizontal resolutions up to 1 km. In particular, we have introduced a new two-dimensional advection-based wind-driven snow redistribution module that is driven by an offline coupling between WindNinja, a wind downscaling model, and Alpine3D. We then show that large accumulation variability can be at least partially explained by terrain-induced wind speed variations which subsequently redistribute snow around rolling topography. By comparing Alpine3D to airborne-derived snow accumulation measurements within a testing domain over Pine Island Glacier in West Antarctica, we demonstrate that our Alpine3D downscaling approach improves surface mass balance estimates when compared to MERRA-2, a global atmospheric reanalysis which we use as atmospheric forcing. In particular, when compared to MERRA-2, Alpine3D reduces simulated surface mass balance root mean squared error by $23.4 \, \mathrm{mm \, w.e. \, yr^{-1}}$ (13%) and increases variance explained by 24%. Despite these improvements, Alpine3D still underestimates observed accumulation variability, thus providing an opportunity for future model improvement.

## 1 Introduction

Ice sheet surface mass balance (SMB) is the difference between mass accumulation and ablation processes at the surface of ice sheets (Lenaerts et al., 2019). Mass accumulation is composed of precipitation as well as condensation and deposition of atmospheric water vapor, whereas ablation processes remove mass from the ice sheet surface via meltwater runoff, sublimation, and evaporation. Additionally, local SMB is influenced by drifting (lowest 2 m of the atmosphere) and blowing (above 2 m) snow, which together we refer to as deposition in the case of net mass gain and erosion in the case of mass loss (Lenaerts and van den Broeke, 2012).

Drifting and blowing snow have been shown to have a substantial effect on Antarctic SMB spatial variability at scales ranging from tens of kilometers (Dattler et al., 2019; Kausch et al., 2020) to meters (Picard et al., 2019). In fact, both Dattler et al. (2019) and Picard et al. (2019) have shown that local deposition can exceed annual precipitation. In addition to redistributing





mass, drifting and blowing snow contribute to mass loss by promoting enhanced sublimation as snow particles are entrained in the lower atmosphere (Palm et al., 2017). In spite of drifting and blowing snow sublimation being a source of mass loss, evaluation of model simulation of these processes remains difficult (Amory et al., 2021). The net effect of drifting and blowing snow is preferential deposition in areas of mass convergence at the expense of areas with net divergence (Lehning et al., 2008). Despite our lack of complete physical understanding of the processes which govern preferential deposition at small spatial

and temporal scales (Comola et al., 2019), deposition and erosion can be conceptually summarized as the local convergence of previously eroded snow particles from upwind minus locally eroded snow (Liston et al., 2007). Local erosion is governed by the direct competition between forces that act to erode snow and those which act to anchor snow to the ice sheet surface. Erosive forces, namely surface wind stress, are controlled by atmospheric boundary layer processes (Paterna et al., 2016), while cohesive forces are controlled by snow-microstructural properties including grain size and bond strength (Clifton et al.,

2006). This interplay between boundary layer and snow-microstructural processes has historically motivated the development of tightly coupled atmospheric and land surface snow models (e.g., Lawrence et al., 2019; Amory et al., 2021; Sharma et al., 2021).

When surface wind stress exceeds cohesive forces at the snow surface, saltation of snow particles is initiated within the lowermost 10 cm of the atmosphere (Pomeroy and Gray, 1990). Within the saltation layer, which is generally assumed to

account for 50 – 75% of wind-driven mass transport (Gromke et al., 2014), atmospheric momentum is entrained by snow particles which eventually collide with and subsequently mobilize additional particles at the snow surface. As particles break apart upon collision, snow grain fragmentation and rounding are observed, resulting in increased density (Vionnet et al., 2012) in a processes referred to as drifting-snow compaction. Reliable model representation of drifting-snow compaction is now particularly attractive owing to recent recent advancements in satellite altimetry technology (e.g. CryoSat-2 and ICESat-2).

Vertical accuracy, spatial resolution, and ground track repeat frequency have all increased, providing precise measurements of ice sheet surface height change (Smith et al., 2020). However, in order to reliably convert these height changes into mass, particularly over short time scales, we rely on accurate snow and firn density estimates from models (The IMBIE team, 2018). Thus, to confidently assign subtle observed changes in height to quantifiable changes in mass, our models must capture the complex spatial and temporal patterns of deposition and erosion.

Current state-of-the-art firn densification models, which are used to convert satellite observed volume changes into mass, successfully capture broad regional variability in firn properties, including density (Ligtenberg et al., 2011; Medley et al., 2020). However, because of the relatively coarse horizontal resolutions at which they are applied (5.5 – 35 km), these models are unable to represent spatial variability in firn processes, including deposition and erosion, at the horizontal scales now sampled by satellites (< 1 km). This spatial gap between satellite observations and firn densification models may not be immediately

important for mass balance retrievals at continental scales (Verjans et al., 2021). Nevertheless, improved model representation of wind-driven snow redistribution at finer spatial scales can be used to constrain regional to local surface mass balance (Rignot et al., 2011), improve volume-to-mass conversions for repeat altimetry (Shepherd et al., 2012; Zwally et al., 2015), provide the ice coring and radar communities with a mechanism to select representative sampling locations for SMB reconstructions (Kausch et al., 2020), and inform future studies by providing baselines to estimate sublimation of drifting and blowing snow.



To facilitate local SMB estimates and reconstructions as well as repeat altimetry interpretation, we present a new technique
for dynamically downscaling Antarctic SMB by building upon the existing Alpine3D v3.3.0 model framework (Section 2.1).
In particular, we describe the use of WindNinja to downscale wind fields onto local topography (Section 2.2), and demonstrate
the use of a one-dimensional snow model to diagnose local erosion (Section 2.3) and then distribute this mass horizontally
across adjacent grid cells using a new, two-dimensional advection-based redistribution module (Section 2.4). We then present
downscaling results at 1 and 3 km horizontal resolution, and evaluate the added value by quantitatively comparing our results,
along with a global SMB product, to a 130 km long airborne radar derived SMB transect in interior West Antarctica over Pine
Island Glacier.

## 2 Methods

### 2.1 Alpine3D: Surface mass balance downscaling framework

We use and further develop the existing Alpine3D v3.3.0 model framework (Lehning et al., 2006) to downscale Antarctic SMB
processes to a target horizontal resolution of up to 1 km. At its core, the model framework exploits the MeteoIO library (Bavay
and Egger, 2014) to handle meteorological preprocessing and downscaling (Section 2.2), SNOWPACK (Bartelt and Lehning,
2002; Lehning et al., 2002b, a), a physics-based land-surface snow model for the detailed description of snow microstructural
properties (Section 2.3), and a new module to calculate horizontal mass fluxes between adjacent grid cells (sections 2.4 and
2.6, Fig. 1).

### 2.2 Atmospheric forcing: Meteorological downscaling

At the snow surface, we prescribe hourly MERRA-2 global atmospheric reanalysis (Gelaro et al., 2017) which we have down-
scaled to the Alpine3D grid using the MeteoIO preprocessing and downscaling library (Bavay and Egger, 2014). These down-
scaled time series include: 2 m air temperature, relative humidity, incoming shortwave and longwave radiation (ISWR and
ILWR), precipitation rate, and 10 m wind speed and direction. 2 m air temperature is first downscaled to the Alpine3D grid
using an ordinary kriging algorithm with a lapse rate of -6 $^{\circ}$C km$^{-1}$ while relative humidity is spatially interpolated according
to Liston and Elder (2006). Precipitation rate and ISWR are then interpolated using inverse distance weighting, with ISWR
undergoing a simple correction for slope and topographic shading (Helbig, 2009). ILWR is interpolated by first calculating the
hourly average of all MERRA-2 grid cells within the domain, and then applying a constant lapse rate of -31.25 W m$^{-2}$ km$^{-1}$.
At the bottom of the firn column, we follow Keenan et al. (2021), by applying the MERRA-2 mean annual surface tempera-
ture as a Dirichlet thermodynamic boundary condition. Note that for all topographic calculations, we use an ICESat-2 derived
digital elevation model (DEM) (Medley et al., 2022).

Because reliable simulations of deposition and erosion require accurate wind speed and direction fields (Reynolds et al.,
2021), we test two approaches for downscaling wind speed and direction from the relatively coarse (0.5° latitude × 0.625°
longitude) MERRA-2 grid. First, we apply the terrain-based index method proposed by Liston et al. (2007) which adjusts





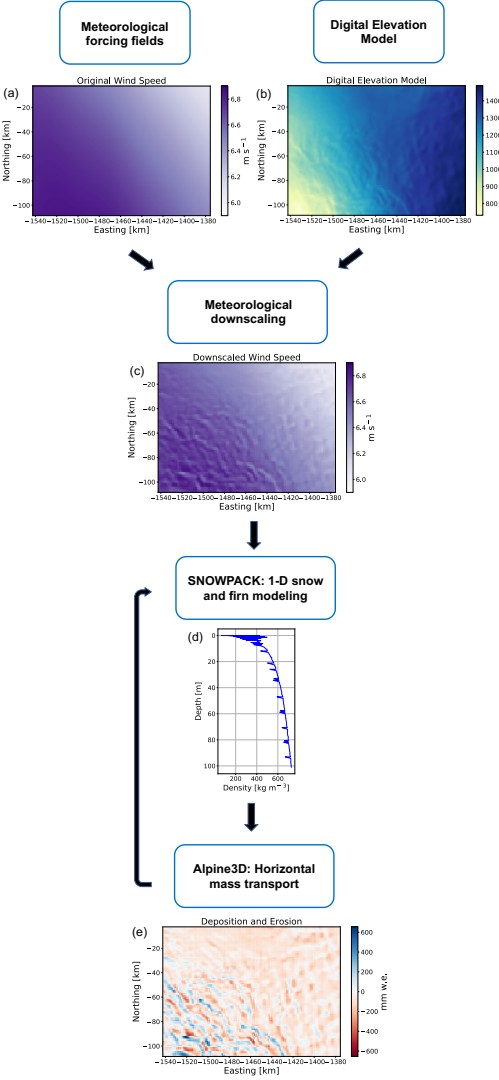

**Figure 1.** Anatomy of an Alpine3D time step: Hourly global atmospheric reanalyis (a) is combined with ICESat-2 derived surface topography (b) to calculate downscaled meteorology (c) at each SNOWPACK model grid cell. The SNOWPACK model is then used to calculate snow microstructural properties at each grid cell (d). Finally, the Alpine3D model is used to calculate horizontal mass transport across adjacent grid cells (e) which is then sent back to SNOWPACK for the next time step.

wind speed and direction based off topographic exposure and sheltering. Note that we use the default slope and curvature weighting factors and choose a topographic length scale of 5 km. However, because Reynolds et al. (2021) showed that simulated snow depth better captured observations when forced with WindNinja downscaled wind fields (Forthofer et al., 2014) compared to the relatively simpler terrain-based index methods presented in Liston et al. (2007), we have implemented

WindNinja (Version 3.7.1) as an alternative offline wind speed and direction downscaling technique within the Alpine3D





modeling framework. WindNinja is a finite element diagnostic model which leverages a mass-conservation solver and DEM to simulate the mechanical effects of terrain on the flow. In terms of WindNinja model configuration, we have selected the "fine" finite element mesh resolution and chosen "grass" as our surface roughness category (snow is not currently an option). Despite representing a significant increase in complexity compared to other terrain-based interpolation techniques (e.g., Liston et al.,

2007), WindNinja is still significantly cheaper than high-resolution numerical weather models (e.g. the Weather Research and Forecasting (WRF) Model), which solve the non-hydrostatic, fully compressible Navier-Stokes equations at multiple vertical levels (Wagenbrenner et al., 2016), and can therefore be run with reasonable computational resources (Section 2.6).

## 2.3 SNOWPACK: One-dimensional physics-based snow model

We use SNOWPACK, a physics-based land-surface snow model, to describe one-dimensional snow and firn processes at each

Alpine3D grid cell. SNOWPACK was originally developed for avalanche warning applications, and has been continuously enhanced in order to represent various cryospheric processes including seasonal snow (Sharma et al., 2021), sea ice snow cover (Wever et al., 2020), and polar firn compaction (Groot Zwaaftink et al., 2013; Keenan et al., 2021). In this study, we use the SNOWPACK physics presented in Keenan et al. (2021), with one exception being a new parameterization for surface roughness length, $z_0$ (m), Eq. (1), tuned to observed seasonal variability between winter and summer surface roughness in

coastal East Antarctica (Amory et al., 2017). In Eq. (1), $\kappa$ is the von Kármán constant (0.4), $T_{2\,\mathrm{m}}$ is the 2 m air temperature (°C), and $C_{\mathrm{DN10}}$ is the neutral drag coefficient at 10 m, Eq. (2).

$$z_0 = \begin{cases} \frac{10}{\exp{\frac{\kappa}{\sqrt{C_{\mathrm{DN10}}}}}} & T_{2\mathrm{m}} > -20\,°C \\ 0.0002 & T_{2\mathrm{m}} \leq -20\,°C \end{cases} \tag{1}$$

$$C_{\mathrm{DN10}} = 2.7 \times 10^{-3} + 9.0 \times 10^{-5} T_{2\mathrm{m}} + 1.5 \times 10^{-6} T_{2\mathrm{m}}^2 \tag{2}$$

In SNOWPACK, snow redistribution is initiated when the friction velocity $u_*$ (m s$^{-1}$) exceeds the surface threshold friction

velocity $u_{*\mathrm{th}}$ (m s$^{-1}$, Eq. (3)), which is calculated diagnostically as a function of snow microstructural properties including snow grain sphericity $SP$ (0 - 1), radius $r_\mathrm{g}$ (m), bond radius $r_\mathrm{b}$ (m), and coordination number $N_3$ (Lehning and Fierz, 2008).

$$u_{*\mathrm{th}} = \sqrt{\frac{A\rho_\mathrm{i} g r_\mathrm{g}(SP + 1) + B\sigma N_3 \frac{r_\mathrm{b}^2}{r_\mathrm{g}^2}}{\rho_\mathrm{a}}} \tag{3}$$

In Eq. (3), $\rho_\mathrm{i}$ is the density of ice (917 kg m$^{-3}$), $\rho_\mathrm{a}$ is the density of air (1.1 kg m$^{-3}$), $g$ is the gravitational acceleration (9.8 m s$^{-2}$), $\sigma$ is a reference shear strength set to 300 $Pa$, while constants $A$ and $B$ are set to 0.02 and 0.0015 respectively.

Once drifting snow is initiated, layers are eroded from the top of the simulated snow cover until the saltation mass flux $\Phi$ (kg m$^{-2}$ s$^{-1}$, Eq. (4)) is satisfied. These eroded layers are then made available to Alpine3D for horizontal redistribution





across adjacent grid cells (Section 2.4). In Eq. (4), $L$ is a characteristic horizontal length scale, which we call the fetch length, over which the saltating particles in $\Phi$ are scaled to calculate local erosion from the firn layer.

$$\Phi = \frac{0.0014\rho_\mathrm{a}u_*(u_* - u_{*\mathrm{th}})(u_* + 7.6u_{*\mathrm{th}} + 205)}{L} \tag{4}$$

In contrast to our previous study (Keenan et al., 2021), we revert $L$ from 10 m back to the original SNOWPACK value of 70 m. We make this choice because Alpine3D simulations with a fetch length of 30 m or less were found to significantly overestimate the magnitude of deposition and erosion when compared to observations (Section 3.6). However, because we are not aware of any direct observations, $L$ can be effectively considered a tuning parameter whose magnitude is inversely proportional to the amount of eroded mass accounted for in the saltation mass flux $\Phi$ (Wever et al., 2022). Note that we do

not explicitly account for blowing snow suspension in our model. We make this pragmatic decision primarily for the three following reasons: 1) full prognostic solution of blowing snow transport via suspension requires wind vector fields at multiple vertical levels (Sharma et al., 2021), 2) few observational case studies quantitatively partition between saltation and suspension-driven transport and their eventual effect on deposition and erosion, and 3) the poorly constrained fetch length $L$ can be tuned such that the saltation mass flux $\Phi$ naively accounts for mass fluxes from both saltation and suspension (Keenan et al., 2021).

## 2.4   Alpine3D: Numerical treatment of deposition and erosion

At each time step, the saltation mass flux $\Phi$ (see Eq. (4)) at each grid cell is scaled to a saltation mass $M_\mathrm{s}$ $(\mathrm{kg\,m^{-2}})$ by multiplying $\Phi$ by the Alpine3D time step $\Delta t_\mathrm{A3D}$ (s) (Eq. (5)).

$$M_\mathrm{s} = \Phi\Delta t_\mathrm{A3D} \tag{5}$$

    $M_\mathrm{s}$ is then made available to Alpine3D for downstream redistribution by the wind, resulting in a local saltation mass pertur-

bation $\Delta M_\mathrm{s}$ $(\mathrm{kg\,m^{-2}})$ which is positive in the case of net deposition and negative in the case of erosion. We calculate $\Delta M_\mathrm{s}$ by treating wind-driven snow redistribution as a two-dimensional horizontal advection problem (Eq. (6)) where $\boldsymbol{u}_\mathbf{s}$ is the saltation velocity vector $(\mathrm{m\,s^{-1}})$. In our implementation, $\boldsymbol{u}_\mathbf{s}$ is defined as parallel to the 10 m wind speed unit vector $\hat{\boldsymbol{U}}_{10\,\mathrm{m}}$ with a magnitude parameterized as a function of $u_{*\mathrm{th}}$ (Eq. (7)) according to Pomeroy and Gray (1990).

$$\frac{\partial M_\mathrm{s}}{\partial t} + \boldsymbol{u}_\mathbf{s} \cdot \nabla M_\mathrm{s} = 0, \tag{6}$$

$\boldsymbol{u}_\mathbf{s} = 2.8u_{*\mathrm{th}}\hat{\boldsymbol{U}}_{10\,\mathrm{m}}$                              (7)

    We solve for $\Delta M_\mathrm{s}$ by numerically integrating Eq. (6) forward in time using a first-order accurate, upwind finite difference scheme with an adaptive sub-time step $\Delta t_\mathrm{CFL}$ (s) in order to ensure numerical stability under the Courant-Friedrichs-Lewy





(CFL) condition (Courant et al., 1928). The local saltation mass perturbation at the $i^{\text{th}}$ sub-time step originating from advection of saltating snow in the $\hat{x}$ and $\hat{y}$ directions $\delta M_{\text{s},x}^{i}$, $\delta M_{\text{s},y}^{i}$ (kg m$^{-2}$, Eqs. (8) and (9)), are then calculated with $\Delta X$ being the Alpine3D horizontal grid resolution.

$$\delta M_{\text{s},x}^{i} = \begin{cases} \frac{\left(M_{\text{s}(x-1,y)} - M_{\text{s}(x,y)}\right)}{\Delta X}(\boldsymbol{u_s} \cdot \hat{x})\Delta t_{\text{CFL}} & (\boldsymbol{u_s} \cdot \hat{x}) > 0 \\ \frac{\left(M_{\text{s}(x,y)} - M_{\text{s}(x+1,y)}\right)}{\Delta X}(\boldsymbol{u_s} \cdot \hat{x})\Delta t_{\text{CFL}} & (\boldsymbol{u_s} \cdot \hat{x}) < 0 \end{cases} \tag{8}$$

$$\delta M_{\text{s},y}^{i} = \begin{cases} \frac{\left(M_{\text{s}(x,y-1)} - M_{\text{s}(x,y)}\right)}{\Delta X}(\boldsymbol{u_s} \cdot \hat{y})\Delta t_{\text{CFL}} & (\boldsymbol{u_s} \cdot \hat{y}) > 0 \\ \frac{\left(M_{\text{s}(x,y)} - M_{\text{s}(x,y+1)}\right)}{\Delta X}(\boldsymbol{u_s} \cdot \hat{y})\Delta t_{\text{CFL}} & (\boldsymbol{u_s} \cdot \hat{y}) < 0 \end{cases} \tag{9}$$

$\Delta t_{\text{CFL}}$ is optimized such that we choose the maximum possible sub-time step while simultaneously satisfying the CFL condition (Eq. (10)).

$$(\mid \boldsymbol{u_s} \cdot \hat{x} \mid + \mid \boldsymbol{u_s} \cdot \hat{y} \mid)\frac{\Delta t_{\text{CFL}}}{\Delta X} < 1 \tag{10}$$

Note that in the case of the final sub-time step, we reduce $\Delta t_{\text{CFL}}$ such that the sum of all $N$ sub-time steps is equal to $\Delta t_{\text{A3D}}$. By summing equations 8 and 9 across all $\Delta t_{\text{CFL}}$ within $\Delta t_{\text{A3D}}$, we are able to finally calculate the local saltation mass perturbation $\Delta M_{\text{s}}$ (kg m$^{-2}$, Eq. (11)) which is ultimately sent to SNOWPACK as a mass flux available at the next time step.

$$\Delta M_{\text{s}} = \sum_{i=1}^{N}(\delta M_{\text{s},x}^{i} + \delta M_{\text{s},y}^{i}) \tag{11}$$

Our upwind finite difference scheme cannot be applied at the edges of a domain, therefore we must prescribe boundary conditions. In this application (Section 2.5) we simulate a finite spatial domain. In order to force our domain to behave more like an infinite domain, we implement periodic boundary conditions (Hames et al., 2021) which implies that deposition and erosion integrate to zero throughout the modeling domain. Furthermore, we note that our model currently makes a significant simplification by not considering sublimation of snow particles actively entrained in the atmosphere above the snow surface.

## 2.5 Pine Island Glacier experimental domain

As a demonstration of our Alpine3D downscaling framework, we define a 198 km $\times$ 137 km modeling domain centered over the Pine Island Glacier basin in West Antarctica (Fig. 2a, red rectangle). In this domain, surface elevations range from approximately 700–1500 m leading to relatively large surface slopes (mean: $0.64°$, 10% quantile: $0.18°$, 90% quantile: $1.3°$) typical of the Antarctic escarpment zone. Strong and steady winds driven by consistent katabatic forcing, combined with rolling topography, have been shown to drive SMB variability of up to a factor of five (200–1000 mm w.e.yr$^{-1}$) over horizontal scales on the order of 10 km (Dattler et al., 2019). In fact, Dattler et al. (2019) found the largest SMB spatial variability among



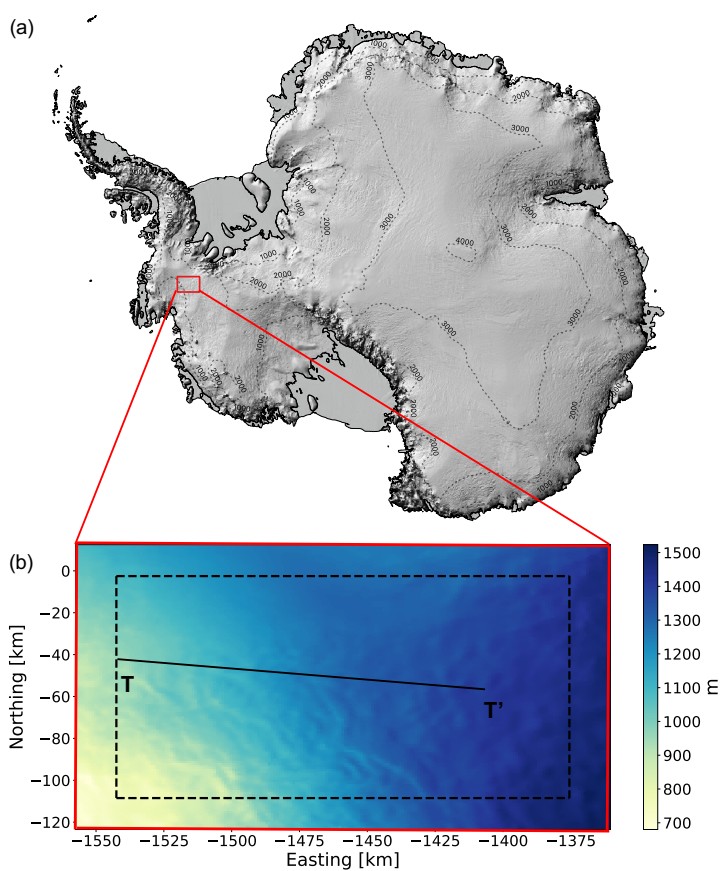

**Figure 2.** Alpine3D modeling and analysis domains in the EPSG 3031 coordinate system: (a) The 198 km × 137 km Alpine3D modeling domain (red rectangle) is centered over the Pine Island Glacier basin in West Antarctica. (b) Surface topography of Alpine3D modeling (red) and analysis (black dashed) domains with the location of the 130 km observed SMB transect (black line, T–T'). Panel (a) was created using the Norwegian Polar Institute's Quantarctica package (Matsuoka et al., 2018).

their 17,500 km of flight line measurements in this domain (Section 2.7). This region therefore provides an opportunity to evaluate our SMB downscaling framework, while still being of modest spatial extent and therefore acceptable computational cost necessary for continuous model development. In terms of model initialization, we borrow from our previous study, Keenan

et al. (2021), by initializing each Alpine3D grid cell with a 100 m thick firn column whose properties are uniquely determined by running one-dimensional SNOWPACK (Section 2.3) with atmospheric forcing taken from the nearest MERRA-2 grid cell (Section 2.2). Note that in order to minimize the relative importance of boundary conditions for drifting snow, upon analysis we remove all grid cells within 15 km of the four domain boundaries, resulting in grid size of 168 km × 107 km (Fig. 2b black dashed rectangle).



## 2.6 Computational parallelization and benchmarking

To enable efficient numerical parallelization, the Alpine3D modeling framework supports a hybrid OpenMP/MPI implementation. For benchmarking, we perform calculations on a 1 km horizontal resolution 27,126 km$^2$ domain (Section 2.5) using general purpose compute nodes with 24 parallel CPUs per node. In the case of offline WindNinja wind speed and direction downscaling, we use 1 node (OpenMP, 24 CPUs) for 1 hour per calendar year, resulting in an efficiency of 1130 km$^2$ yr (CPU hr)$^{-1}$. Whereas for SNOWPACK and Alpine3D caclulations, we leverage 4 nodes (hybrid OpenMP/MPI, 96 total CPUs) for 18 hours per calendar year, netting an efficiency of 16 km$^2$ yr (CPU hr)$^{-1}$. WindNinja is therefore at least one order of magnitude cheaper than combined SNOWPACK and Alpine3D, meaning that its added complexity is justified if downscaled SMB estimates are significantly improved.

## 2.7 Surface mass balance observations

To evaluate our SMB downscaling, we compare with a 130 km long, radar-derived accumulation transect (Fig. 2b black line T–T') developed by Dattler et al. (2019). This annual-averaged snow accumulation product captures spatial variability in along-track accumulation by tracking fluctuations in radar-derived firn isochrone thickness. Horizontal variations in thickness are then converted to mass using the Herron and Langway (1980) firn density model. At 25 km horizontal resolution, Dattler et al. (2019) set the mean accumulation to that of MERRA-2 atmospheric reanalysis. While at smaller scales, accumulation is modified by the combination of firn isochrone thickness and simulated firn density.

## 3 Results and Discussion

### 3.1 Impact of wind downscaling method on simulated surface mass balance

To demonstrate the differences between downscaling techniques, we have downscaled MERRA-2 winds from the year 2015 using both the Liston and WindNinja (Section 2.2) algorithms to the 1 km Alpine3D grid (Fig. 3). WindNinja predicts a higher average wind speed than Liston (6.5 m s$^{-1}$ vs. 6.1 m s$^{-1}$) and is more consistent with MERRA-2 (6.6 m s$^{-1}$), which can be explained by WindNinja's mass conservation solver and Liston's imposed terrain weighting factor of 0.5–1.5 times the original interpolated value (Reynolds et al., 2021) . Notably, both downscaling techniques predict local topography-driven wind speed variability not captured by MERRA-2, however this variability is in opposite phase, meaning that when WindNinja predicts relatively high wind speed, Liston predicts locally lower wind speed. Furthermore, WindNinja predicts 2–3 times larger local topography-driven wind speed variability than Liston, leading us to expect larger SMB variability in Alpine3D simulations forced with WindNinja winds compared to Liston-driven simulations. Likewise, because predicted wind speed accelerations are out of phase between the WindNinja and Liston downscaling methods, we anticipate the subsequent offline coupling with Alpine3D to predict opposite patterns of deposition and erosion. Note that our modeling domain (Section 2.5) lacks the necessary wind speed and direction observations required for a robust evaluation, therefore we focus our evaluation





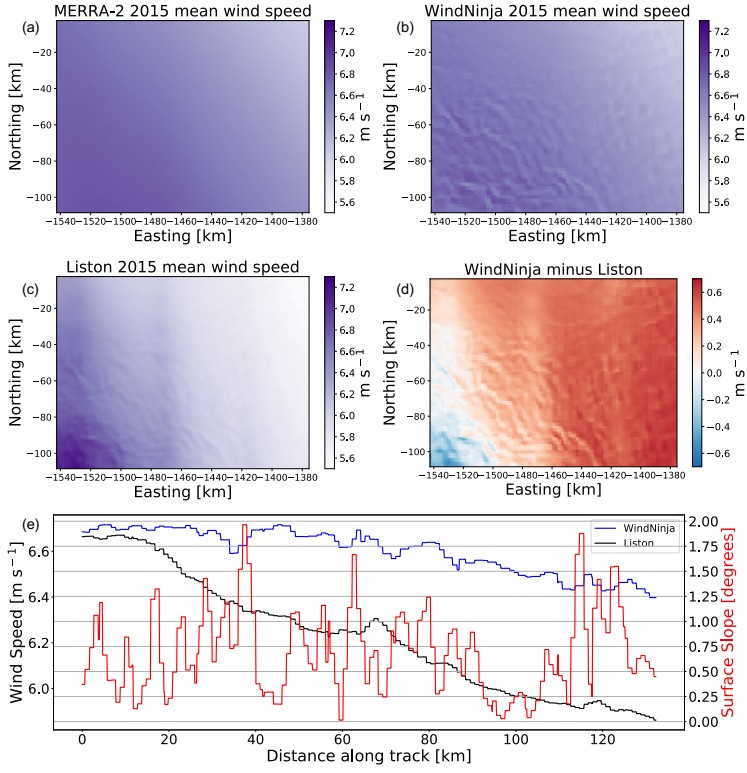

**Figure 3.** Example of wind speed downscaling for the year 2015: (a) Annual mean MERRA-2, (b) WindNinja, and (c) Liston 10 m wind speed. (d) Annual mean WindNinja minus Liston wind speed. (e) Transect (T–T' in Fig. 2b) of annual mean WindNinja and Liston wind speed as well as transect surface slope.

on downscaled SMB, which ultimately integrates wind speed and direction and can be evaluated against observations (Section 2.7).

Owing to the largely contrasting results between the WindNinja and Liston wind downscaling techniques, we have performed two 2015 Alpine3D simulations, one using WindNinja and one using Liston winds. As discussed above, the WindNinja driven Alpine3D simulation exhibits larger SMB variability than the Liston driven simulation (Fig. 4) because of larger wind speed variability in the former. Additionally, the WindNinja driven Alpine3D simulation predicts SMB variability in phase with observations (slope = 0.26, p < 0.001), whereas the Liston driven simulation predicts SMB variability out of phase with observations (slope = -0.12, p < 0.001). Because the offline coupling between the Liston downscaling algorithm and Alpine3D predicts SMB variability out of phase with observations, we determine that the Liston algorithm is not suitable for our application. Thus, moving forward we only consider Alpine3D simulations driven by an offline coupling with WindNinja.

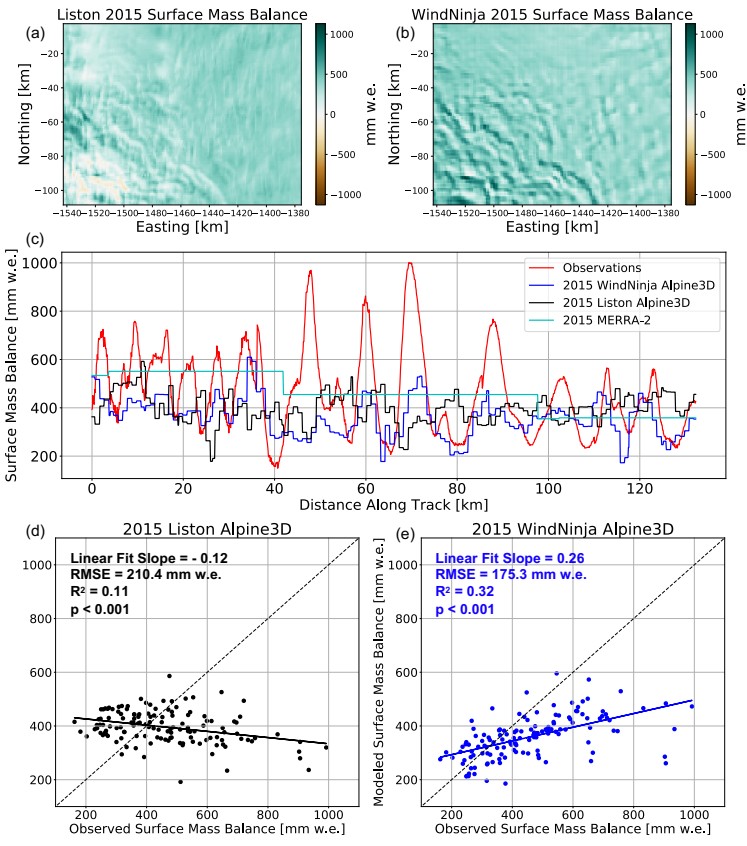

**Figure 4.** Surface mass balance downscaling: (a) 2015 Liston and (b) WindNinja Alpine3D SMB. (c) Transect of observed (red), 2015 WindNinja Alpine3D (blue), Liston Alpine3D (black) and MERRA-2 (cyan) SMB. (d, e) Scatter plot of (d) 2015 Liston and (e) WindNinja Alpine3D SMB vs. observations.

## 3.2  Alpine3D downscaled surface mass balance

Given the demonstrated performance advantage of using the offline WindNinja coupling with Alpine3D (Section 3.1), we further this analysis by evaluating a six year long Alpine3D simulation using WindNinja (2015–2020). By comparing with an approximately 130 km long observational transect, we found that based on the simulated 2015–2020 mean annual SMB, Alpine3D correctly predicts the relative locations of peaks and troughs in observed SMB and explains an additional 24% variance explained compared to MERRA-2 (31% (17 − 47% 95% confidence interval) vs. 7% (2 − 15% 95% confidence interval), Fig. 5). In particular, Alpine3D appears to improve over MERRA-2 SMB predictions as indicated by a reduction in root mean squared error (152.7 mm w.e. yr$^{-1}$ vs. 176.1 mm w.e. yr$^{-1}$). Despite improving over MERRA-2 (slope = 0.13, p = 0.003), Alpine3D still underestimates the magnitude of observed SMB variability, as indicated by the linear fit slope being less than 1 (slope = 0.25, p < 0.001).

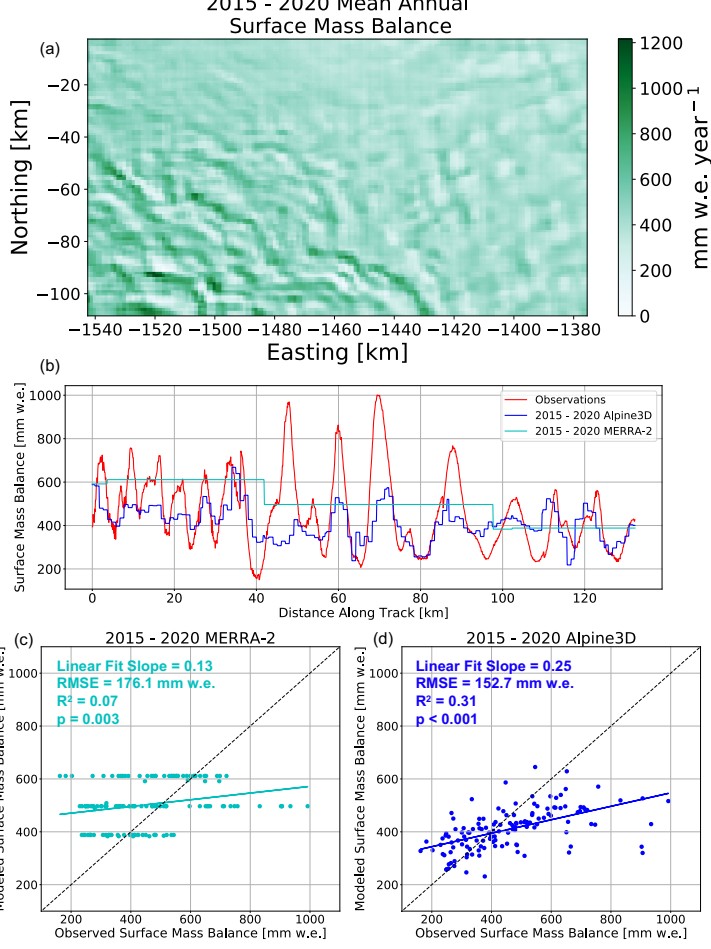

**Figure 5.** Surface mass balance downscaling: (a) 2015 – 2020 Alpine3D SMB. (b) Transect of observed (red), 2015 – 2020 mean Alpine3D (blue) and MERRA-2 (cyan) SMB. (c, d) Scatter plot of (c) 2015 – 2020 MERRA-2 and (d) Alpine3D SMB vs. observations.

Generally speaking, Alpine3D correctly predicts the location of deposition and erosion as evidenced by the relative positioning of peaks and troughs in SMB. However, the simulated amplitude of deposition and erosion is underestimated when compared to observations. This underestimation could potentially be ameliorated by dereasing the fetch length and therefore increasing the saltation mass flux, although we show in Section 3.6 that doing so results in locally excessive erosion and in fact, negative SMB. While most peaks and troughs in SMB are captured by Alpine3D, some are missed, for example near 45 km on the transect in Fig. 5b. Several processes likely contribute to this significant model error, however the primary driver is a lack of terrain-driven wind acceleration at this point of the transect (Fig. 3e). Despite indications that this area has some of the largest SMB variability within the transect, surface slopes in the DEM are comparatively modest ($< 1.25°$, Fig 3). Without significant surface slopes, the downscaled wind speeds are relatively homogeneous, leading Alpine3D to miss the corresponding





observed SMB variability. This relatively homogeneous wind field in the presence of large observed accumulation gradients,
can be potentially attributed to a combination of terrain representation errors in the DEM as well as process representation
errors and simplifications within the WindNinja downscaling algorithms. Furthermore, it is worth noting that over the length of
the transect, MERRA-2 2015–2020 mean annual SMB exceeds that of observations (504 and 461 $\mathrm{mm\,w.e.\,yr^{-1}}$), indicating
that the 2015–2020 period exceeded the long term average. Despite this, mean Alpine3D downscaled SMB over the transect
(410 $\mathrm{mm\,w.e.\,yr^{-1}}$) is less than both observations and MERRA-2. This discrepancy between MERRA-2 and Alpine3D is
245 explained by net divergence of saltating snow out of the analysis domain (Fig. 2). Because the observations are deliberately
forced to have the same mean as MERRA-2 (Section 2.7), we are unable to properly evaluate this predicted discrepancy against
measurements.

### 3.3 Link between topography, surface winds, and drifting snow redistribution

Intuitively, Alpine3D is designed to preferentially erode snow in wind exposed areas and redeposit it downwind in wind
sheltered zones. This is primarily achieved through the offline coupling between WindNinja and Alpine3D. In areas with
spatially varying surface slope, WindNinja adjusts wind speed and direction to account for mechanical effects of terrain on
the flow (Fig. 6). In the case of the transect (location shown in Fig. 2), surface slopes are relatively modest, but regularly
vary between 0.25–2.0°. Ultimately, this rolling terrain leads WindNinja to simulate equally subtle wind speed variations up
to approximately 0.1 $\mathrm{m\,s^{-1}}$. The end result is enhanced erosion (0–250 $\mathrm{mm\,w.e.\,yr^{-1}}$) of snow in relatively windy areas and
255 subsequent deposition (up to 125 $\mathrm{mm\,w.e.\,yr^{-1}}$) in less windy areas. Expanding this analysis to our entire domain, we see
that rolling topography drives spatially variable patterns of deposition and erosion (Fig. 6d). On average wind-driven transport
processes remove 50 $\mathrm{mm\,w.e.\,yr^{-1}}$ from the analysis domain, while local values of erosion and deposition range from -
514 – 690 $\mathrm{mm\,w.e.\,yr^{-1}}$. Overall, deposition and erosion processes modify snow accumulation with an average magnitude of
89 $\mathrm{mm\,w.e.\,yr^{-1}}$, which corresponds to 23% of mean annual accumulation.

### 260 3.4 Link between density and deposition and erosion

Altimetry-based ice sheet mass balance retrievals rely on firn density products to perform volume-to-mass conversions. Thus,
accurate firn models are necessary for reliable local, regional, and continental scale mass balance assessments. Because erosion
exposes older and denser snow layers while also coinciding with areas of elevated wind speed, one could plausibly expect
Alpine3D to predict areas with net erosion to coincide with relatively high surface density. Interestingly, we show that this is
265 not the case. Instead, Alpine3D predicts wind-driven compaction to be the dominant process, whereby eroded snow particles
experience densification via fragmentation and rounding on their way to subsequent deposition (Vionnet et al., 2012). By
comparing 2015–2020 mean annual Alpine3D surface density (defined as the mean of the top 10 cm) with deposition and
erosion, we find that the two are highly correlated (R = 0.74, Fig. 7). Although this positive correlation is not directly verified
by in-situ observations, Grima et al. (2014) found a similar correlation between surface density and local SMB perturbations
over Thwaites Glacier. If for the moment we assume this model result is correct, then the accumulation product compiled by
Dattler et al. (2019) most likely underestimates SMB spatial variability because of the constant surface density assumption



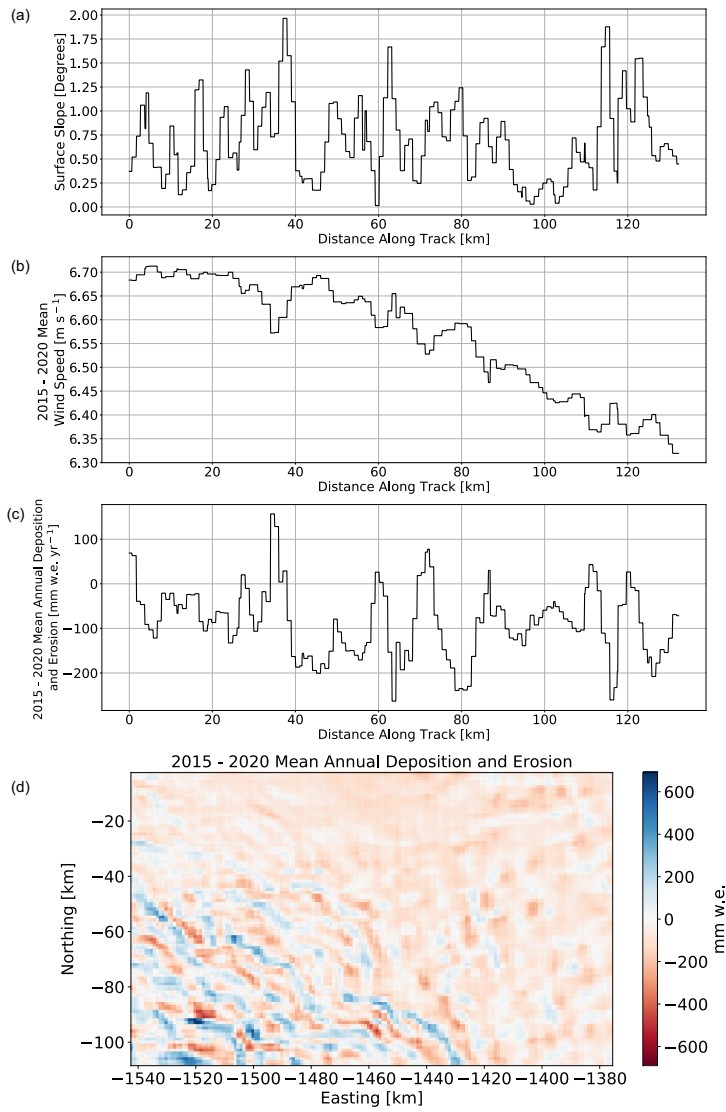

**Figure 6.** Connection between topography and snow redistribution. (a) Transect of surface slope, (b) 2015–2020 mean 10 m wind speed, and (c) 2015–2020 mean deposition and erosion. (d) Map of 2015–2020 mean annual deposition and erosion.

used to initialize their firn model. Likewise, in the case of repeat altimetry, we can deduce that variable surface height change in the vicinity of rolling topography may represent enhanced mass change variability than we might otherwise expect with a homogeneous density field. Overall, these findings should be interpreted with caution as they are not yet directly verified by observations, but likewise warrant the consideration of detailed firn density products when interpreting airborne snow accumulation radar and satellite altimetry.


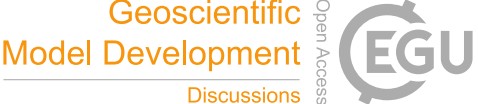


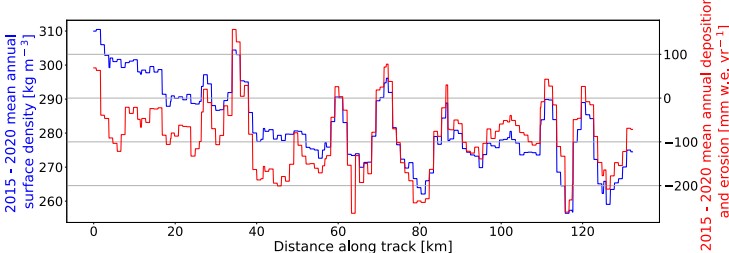

**Figure 7.** Connection between deposition and erosion and surface density (defined as the mean of the top 10 cm). Transect of 2015–2020 mean Alpine3D surface density (blue) and annual deposition and erosion (red).

### 3.5 Impact of horizontal model resolution on downscaled SMB

Depending on the Alpine3D downscaling application, it may be desirable to decrease horizontal resolution in favor of larger domains or longer simulations. We also want to test the sensitivity of the numerics used in Alpine3D to the spatial resolution. In order to inform future modeling studies which will inevitably weigh computational costs against spatial and temporal detail, here we describe results from a 1 and 3 km horizontal resolution 2015 Alpine3D simulation. Owing primarily to reduced number of individual Alpine3D grid cells, the 3 km simulation requires only 10% of the 1 km simulation's computational resources (Section 2.6).

Generally speaking, both simulations capture observed topographic-driven variability in SMB (Fig. 8), however the 1 km simulation predicts 40% larger SMB variability than the 3 km simulation. This finding can be explained by the relatively smoother terrain in the 3 km simulation, and consequently reduced terrain-induced wind speed variability. Reduced wind speed variations across subtle topography ultimately reduced the magnitude of deposition and erosion leading to a more homogeneous SMB field (Fig. 8a vs. Fig. 8b). That said, the 3 km simulation explains an additional 5% variance of observed accumulation variability compared to the 1 km simulation, all while reducing the RMSE by 8 mm w.e. yr$^{-1}$ (Table 1). These improved statistics could be explained by reduced wind-transport divergence in the 3 km simulation (-48.8 vs. -51.4 mm w.e. yr$^{-1}$) and subsequent reduction of the mean bias. However, given that the observed accumulation mean is controlled by MERRA-2 (Section 2.7), caution should be taken in interpreting the absolute value of observations. Overall, these findings suggest that the 1 and 3 km simulations perform similarly, implying that the Alpine3D framework can be applied at horizontal resolutions greater than 1 km without unacceptable degradation of performance. Furthermore, because performance at 3 km horizontal resolution is comparable to that of 1 km, future Alpine3D applications can consider increasing grid spacing up to 3 km in order to expand the spatial and temporal bounds of their simulation.

### 3.6 Sensitivity to prescribed fetch length

As indicated in Section 2.3, the SNOWPACK fetch length is an uncertain tuning parameter that controls the magnitude of the saltation mass flux. Furthermore, because no suitable observations or theoretical framework firmly suggest which value of fetch





**Table 1.** Linear regression statistics for different choice of horizontal resolution. Coefficient of determination ($R^2$), root mean squared error (RMSE), and linear regression slope for 2015 Alpine3D simulations with 1 and 3 km horizontal resolution against observations.

| Alpine3D Horizontal Resolution | 1 km | 3 km |
|---|---|---|
| $R^2$ | 0.32 | 0.37 |
| RMSE (mm w.e. yr$^{-1}$) | 175 | 167 |
| Linear regression slope | 0.25 | 0.18 |

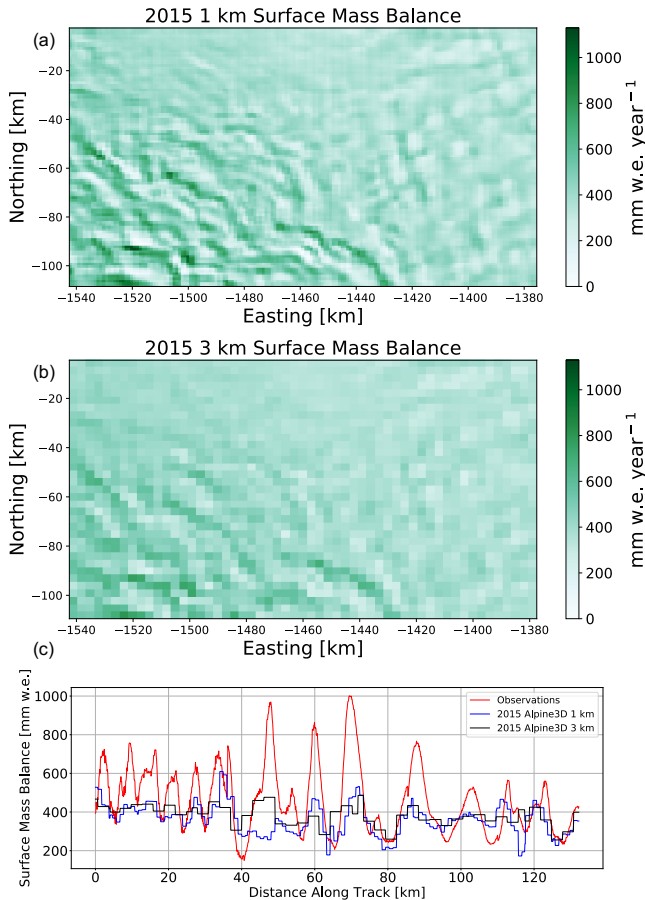

**Figure 8.** Connection between horizontal resolution and simulated SMB. 2015 simulated SMB with (a) 1 and (b) 3 km horizontal resolution. (c) Transect of observed (red) and simulated SMB with 1 (blue) and 3 km (black) horizontal resolution.

length to choose, here we present results from a sensitivity study designed to quantify the effect of varying the fetch length between 30, 70, and 110 m. As expected, progressively increasing the fetch length from 30 to 110 m, decreases the magnitude of deposition and erosion and consequently the spatial variability of 2015 Alpine3D simulated surface mass balance (Fig. 9).





Note that by decreasing the fetch length, we increase the simulated SMB variability to be more in line with observations. However, decreasing the fetch length to 30 m, results in unrealistically strong erosion, as indicated by negative SMB values in 4.2% of grid cells. In fact, the 30 m fetch length simulation produces a larger RMSE than both the 70 and 110 m simulations (273, 175, and 164 mm w.e. yr$^{-1}$ respectively, Table 2). Wever et al. (2022) showed that stand-alone SNOWPACK simulation results were very similar for the range of fetch lengths tested here (30, 70, and 110 m), suggesting that the impact we find in Alpine3D is primarily a result of the snow redistribution module (Section 2.4).

Despite the relatively limited breadth of our fetch length ensemble, we observe a notable trade off. As we decrease fetch length, simulated SMB variability approaches that of observations (as indicated by linear regression slopes in Table 2) while also increasing typical model errors (RMSE in Table 2). We have therefore identified a potential shortcoming of our Alpine3D downscaling framework, namely that optimizing the fetch length parameter does not result in uniformly improved results. For purely pragmatic reasons, in this study we retain the fetch length approach by choosing a middle ground value of 70 m. Nevertheless, in future model development exercises it would be useful to drop this reliance on fetch length, and instead focus on diagnosing the locally eroded mass purely from simulated snow characteristics and surface wind stress.

**Table 2.** Linear regression statistics for different choice of fetch length. Coefficient of determination ($R^2$), root mean squared error (RMSE), and linear regression slope for 2015 Alpine3D simulations with 30, 70, and 110 m fetch length against observations.

| Fetch Length | 30 m | 70 m | 110 m |
|---|---|---|---|
| $R^2$ | 0.30 | 0.32 | 0.31 |
| RMSE (mm w.e. yr$^{-1}$) | 273 | 175 | 164 |
| Linear regression slope | 0.55 | 0.25 | 0.17 |

## 4 Conclusions

The primary way ice sheets gain mass is through snow accumulation at the ice sheet surface, referred to as SMB. Throughout the last two decades, regional climate models have been progressively developed to simulate large-scale ice sheet SMB fields. However, recent advancements in observational capabilities (e.g. repeat satellite altimetry and airborne snow radar) indicate that SMB varies predictably in the presence of rolling topography. Despite the importance of capturing this SMB variability for assigning a mass change to repeat altimetry measurements, regional climate models are not capable of resolving this variability. To begin to close the gap between observations and existing regional climate models, and therefore improve local SMB estimates and support interpretation of repeat altimetry, we have expanded upon the Alpine3D framework in order to dynamically downscale Antarctic SMB.

In our implementation of Alpine3D, we leverage MeteoIO and WindNinja software to downscale meteorology onto the native Alpine3D grid. At each grid cell, the SNOWPACK snow model is used to diagnose snow properties including, density, temperature, grain size, and other snow microstructure parameters. With this information, SNOWPACK calculates a saltation mass flux for each time step and then sends this information to Alpine3D, where horizontal snow redistribution is treated



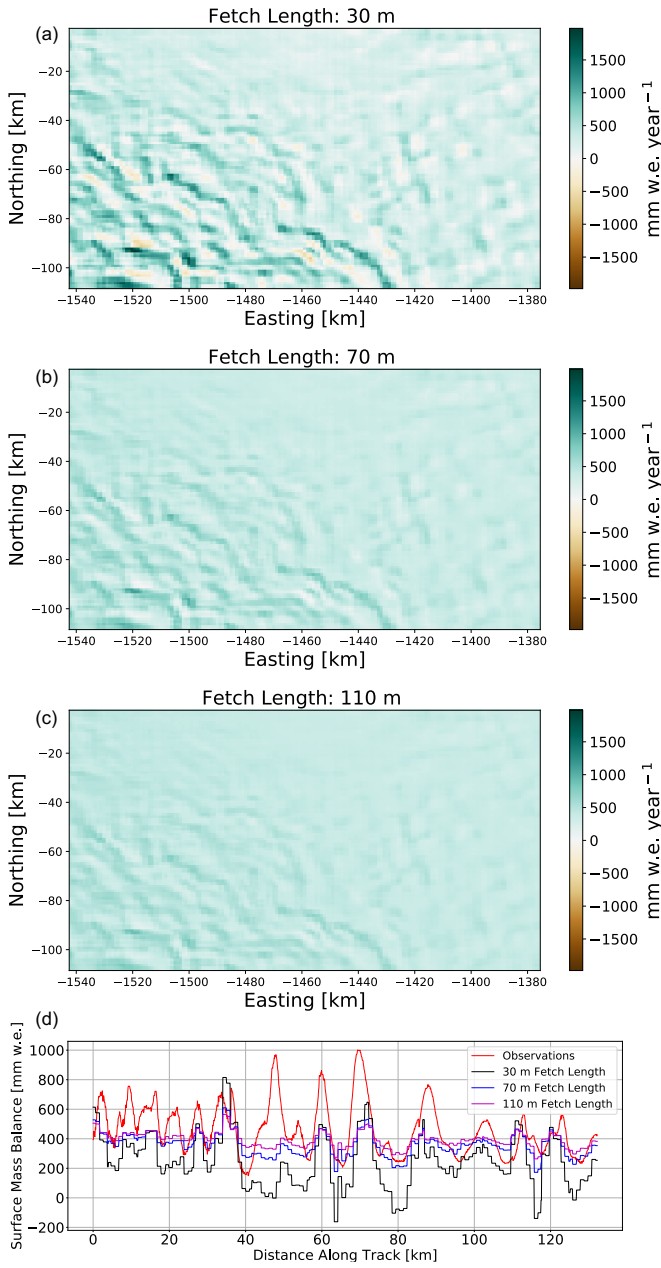

**Figure 9.** Connection between prescribed fetch length and simulated SMB. 2015 simulated SMB with (a) 30, (b) 70, and (c) 110 m fetch length. (d) Transect of observed (red), and simulated SMB with 30 (black), 70 (blue), and 110 m (magenta) fetch length.

as a two-dimensional advection problem. WindNinja downscaling results suggest that rolling topography with surface slopes
between 0.25 - 2.0° are responsible for subtle terrain induced mechanical effects that vary mean wind speeds by $0.1 \, \mathrm{m \, s^{-1}}$.





These wind speed perturbations ultimately drive spatially variable patters of deposition and erosion whose mean magnitude corresponds to 23% of mean annual accumulation.

By comparing 2015 – 2020 mean annual SMB downscaling results over a portion of Pine Island Glacier, West Antarctica to a 130 km long observational SMB transect, we found that, when compared to MERRA-2, Alpine3D reduces typical SMB errors by 23.4 mm w.e. (13%) and increases variance explained by 24%. Furthermore, we show that Alpine3D produces a similarly skillful downscaled SMB product when run at 3 km horizontal resolution. This suggests that Alpine3D can be run at a coarser resolution than what is primarily shown in this study (1 km horizontal resolution), therefore considerably reducing computational cost. Finally, despite the promising downscaling results presented in this study, Alpine3D still relies on the fetch length tuning parameter to scale the saltation mass flux. We have shown the magnitude of deposition and erosion, and consequently SMB, are sensitive to the choice of fetch length. Because it is at the very least difficult, and quite likely impossible, to uniformly optimize the fetch length, we suggest that future studies migrate towards diagnosing the saltation mass flux purely from known quantities including snow properties and surface wind stress.

*Code and data availability.* MeteoIO, SNOWPACK and Alpine3D are software published under the GNU LGPLv3 license by the WSL Institute for Snow and Avalanche Research SLF, Davos, Switzerland at https://gitlabext.wsl.ch/snow-models. The repository used to develop the versions of MeteoIO, SNOWPACK and Alpine3D used in this study can be accessed at https://github.com/snowpack-model/snowpack with the exact version corresponding to commit f023b9f archived at https://doi.org/10.5281/zenodo.5914787. MERRA-2 atmospheric reanalysis is available at https://gmao.gsfc.nasa.gov/reanalysis/MERRA-2/ and can be retrieved and processed using our workflow available at https://github.com/EricKeenan/download_MERRA2 and archived at https://doi.org/10.5281/zenodo.4560825. WindNinja source code is hosted at https://github.com/firelab/windninja while the version used in this study is permanently archived at https://doi.org/10.5281/zenodo.4474633. Software used to produce the numerical simulations, analyses, and figures presented in this study are archived at https://doi.org/10.5281/zenodo.5914751 and https://doi.org/10.5281/zenodo.5914727. Airborne snow accumulation radar data used in this study are archived at https://doi.org/10.5281/zenodo.3534315.

*Author contributions.* All authors contributed to experimental design, analysis, and manuscript preparation. Eric Keenan carried out Alpine3D simulations, performed the analysis, and produced the figures. Eric Keenan and Nander Wever both contributed to Alpine3D model development.

*Competing interests.* The authors declare that they have no conflict of interest.

*Acknowledgements.* Eric Keenan, Nander Wever, Jan T. M. Lenaerts, and Brooke Medley acknowledge support from the National Aeronautics and Space Administration (NASA), grant 80NSSC18K0201 (ROSES-2016: studies with ICESat-2 and CryoSat-2). This work utilized





the RMACC Summit supercomputer, which is supported by the National Science Foundation (awards ACI-1532235 and ACI-1532236), the

University of Colorado Boulder, and Colorado State University. The Summit supercomputer is a joint effort of the University of Colorado Boulder and Colorado State University. Data storage supported by the University of Colorado Boulder "PetaLibrary".





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
