# Peer review of "A wind-driven snow redistribution module for Alpine3D v3.3.0: Adaptations designed for downscaling ice sheet surface mass balance"

_Geoscientific Model Development, 2022_

## Author Comment (AC1)

Thank you to editor Fabien Maussion as well as Charles Amory and the other, anonymous, reviewer for donating their time and energy to our manuscript. In order to address all comments, we have responded to each of them individually in blue. Line numbers in our responses refer to the track-changed version of the revised manuscript. Some newly added citations in our responses can be found in the revised manuscript.

We apologize for the very long delay in responding to the reviewers comments and revising the manuscript. We thank the editor, reviewers and editorial office for their patience and understanding that this is the unfortunate result of the first, second and last author having transitioned between jobs, as well as moving, in the last year.

On behalf all authors,

Eric Keenan

**General comments:**

The article "A wind-driven snow redistribution module for Alpine3D v3.3.0: Adaptations designed for downscaling ice sheet surface mass balance" presents a strategy for downscaling large scale surface mass balance (SMB) predictions using Alpine3D. Alpine3D is a 3-D model that computes the mass and energy balances of snow covered regions by solving the 1-D snow model SNOWPACK at each grid cell. In the proposed methodology, the meteorological conditions are extracted from MERRA-2 and downscaled to the Alpine3D grid. In addition, snow drift events are modeled with a new 2-D advection scheme that takes into account a parameterization for the mass flux in saltation previously implemented in SNOWPACK. In order to correctly estimate snow drift, high resolution wind fields are needed. They are computed offline with the software WindNinja, which takes into account the small scale topographic features through a digital elevation model (DEM). The proposed approach has the potential to improve our understanding of SMB variability at small scales and can easily be applied to other locations. Even though the snow drift model needs further improvement, the coupling of MERRA-2 outputs, WindNinja wind fields and a snow drift scheme with Alpine3D has significant scientific value. In addition, from the comparison between Alpine3D and the measured annual-averaged snow accumulation over a 130 km transect, the authors show the importance of wind redistribution of snow to the local SMB. However, I think the manuscript can be improved, both from a strategic and scientific points of view. Besides the scientific comments presented below I have one general comment:

1) The article risks promising more than it gives regarding the snow drift model. Emphasis is given to snow drift in the title, in the abstract and in the introduction. However, even though the treatment of erosion and deposition presented in section 2.4 can be considered new, it is highly dependent on the parameterizations for the fluid threshold friction velocity and the mass flux in saltation (eqs. 3 and 4). In particular, it is shown and stated by the authors that equation 4 is highly uncertain, as it relies on a poorly constrained parameter. These equations are standard in SNOWPACK and no improvement is suggested by the authors. In addition, it is not clearly stated why this snow drift model is better than the one previously implemented in Alpine3D (Lehning et al. 2006). In this way, I would suggest counter-balancing the focus on the snow drift model with an extended description of the peripheral developments that are of the utmost importance for a successful downscaling: the downscaling of MERRA-2 meteorological forcing to the Alpine3D grid, the use of WindNinja with the ICESat-2 DEM, and the coupling of WindNinja to Alpine3D. In my view, the technical details of

these contributions are of interest to the users of Alpine3D or other models alike. In addition, it is aligned with the scope of the GMD journal.

We thank the reviewer for this feedback and we agree that indeed some more context would be helpful to include for the reader. First, we would like to mention that we consider redeveloping the saltation mass flux parameterizations, as the one we used in Eq. 4, to be out of scope for this study. Particularly since we have achieved satisfying results using this parameterization in earlier work (Keenan et al., 2021, Wever et al., 2022). We consider the study presented in this manuscript as a 2D expansion of the 1D modeling performed in those studies, which we think justifies leaving the current saltation mass flux calculations in SNOWPACK untouched. Furthermore, it is important to note that the snow drift module as used before by for example Lehning et al. (2006), Mott et al. (2008), and Groot-Zwaaftink et al. (2013b), requires 3D wind speed fields at high temporal resolution, which would require a weather model downscaling tool, which adds substantial complexity to such a study. The fully 3D model is also computationally intensive, as it currently is not parallelized. This means that earlier model studies were restricted to shorter time periods (i.e., simulating a winter season, or a case study) and/or relatively small domain. Finally, it is more difficult to maintain mass balance in the current implementation of the 3D model, since snow and precipitation may remain in suspension and never reach the surface, while erosion is restricted to one layer per time step for code simplicity. Solving the fully 3D suspension, considering the added computational effort, is also not guaranteed to provide much better agreement, as discrepancies with observed snow accumulations have been reported (e.g., Mott et al., 2008, Mott et al., 2010, Gerber et al., 2017), such that we think it is justified to test simpler approaches like the one in our study. In our approach, we avoid these computationally challenging issues. Also note here that the full 3D snowdrift model in fact calls the exact same snow drift functions in the SNOWPACK model to determine drifting snow mass flux as we are applying in this study. We now provide more explanation in the manuscript (see L83-87).

**Specific comments:**

l.13-14: Taking into account the focus that is given in the Conclusions regarding the effect of the parameter L, I suggest moving the focus in this sentence from the underestimation of SMB variability to the sensitivity of the snow accumulation patterns to the saltation model employed.

Excellent point. We have revised to (see L14-17):

"Despite these improvements, our results also demonstrate that considerable uncertainty stems from the employed saltation model, confounding simulations of surface mass balance variability."

l.19-21: The terms drifting and blowing snow are used both to define the processes of aeolion snow transport and the particles aloft. This paragraph focuses on processes (precipitation, sublimation, etc). Hence, I suggest rephrasing so that drifting and blowing snow are presented as processes. An example is provided to clarify the comment made: "Additionally, local SMB is influenced by wind redistribution of snow. This process is generally defined as drifitng snow (when the snow particles are transported by the wind in the first 2 m above the snow surface) or blowing snow (when the snow particles are transported by the wind at greater heights - above 2 m height). We refer to deposition when drifting and blowing snow lead to net mass gain and to erosion when they lead to mass loss."

In retrospect, we agree that our description was rather clunky. We have revised to the following (see L22-27):

"Additionally, local SMB is influenced by the process of wind-driven snow redistribution, which we refer to as deposition in the case of local mass gain and erosion in the case of mass loss (Lenaerts and van den Broeke, 2012). Wind-driven snow redistribution is generally divided into two categories, drifting snow, where snow particles are transported by the wind in the lowermost 2 m of the atmosphere, and blowing snow where snow is redistributed at heights greater than 2 m."

l.34-35: More recent works can be cited describing the effect of interparticle cohesion not only on the fluid threshold but in the whole saltation dynamics (e.g. Comola et al. 2019, Melo et al. 2022).

Thanks for bringing these studies to our attention. Indeed they both provide recent insight into saltation dynamics and snow particle cohesion. Thus we have added their citations to the corresponding sentence (see L40-41).

l.39: I suggest citing also the early work of Schmidt (1980) on the impact of interparticle ice bonds on the fluid threshold.

Thanks for the comment. We unintentionally neglected a discussion of impact forces on the development and sustainment of saltation. We've revised the mentioned sentence to the following (see L44-46):

When the combined effect of surface wind stress and impact force from saltating particles exceeds cohesive forces at the snow surface, saltation of snow particles is initiated or maintained within the lowermost 10 cm of the atmosphere (Schmidt, 1980; Pomeroy and Gray, 1990).

l.42: The work of Amory et al. (2021) can also be cited here - a parameterization for drifting snow compaction is also proposed in their work.

Good point. We have added this reference (see L62).

l.85-86: Is this a standard assumption? Maybe the authors can clarify its validity.

We are not sure if this assumption could really pass with the qualifier "standard", but it has been noted in literature before. For example, Fig. 1.3 in Ligtenberg (2014) shows the dissipation of seasonal temperature fluctuations in the uppermost 10m of the firn, to which the author makes the following remark: "... and the local temperature is equal to the long-term average surface temperature." As we already noted in the manuscript, we also applied this boundary condition in our previous study (Keenan et al., 2021). Furthermore, Alley and Koci (1990) note that "At dry-snow sites such as GISP2, the temperature at 10 m depth typically is within a few tenths of a degree of the mean annual air temperature, with the firn usually colder than the air; however, the difference can be as large as a few degrees". From this, we believe our boundary condition is a reasonable simplification, particularly in the absence of a clear alternative approach, without having the need to simulate the full firn column.

In order to provide more justification for this assumption, we have revised the text to (see L112-116):

"We follow Keenan et al., 2021, by applying the MERRA-2 mean annual surface temperature as a Dirichlet thermodynamic boundary condition at the bottom of the firn column. This assumption is supported by observations from the dry snow zone of the Greenland ice sheet, where differences between mean annual air temperature and firn temperature at 10m were found to be typically within a few tenths of a degree (Alley and Koci, 1990). Ligtenberg (2014) also shows in a model result that the seasonal cycle in firn temperature disappears around 10m depth."

l.121: Even though $\Phi$ has units of mass flux (kg/m2/s), it can only be considered a mass flux if the mass rate of saltating particles per unit width, Q (kg/m/s), is assumed to be deposited along a fetch of L meters long. From my point of view, only in this way it makes sense to describe $\Phi$ as the mass rate of particles "crossing" the section Ly times Lx, where Ly is the width and Lx is the fetch length L. Is this the meaning of L? Even though this parameter is not well constrained, I think an effort should be made to better define it.

"Even though $\Phi$ has units of mass flux (kg/m2/s), it can only be considered a mass flux if the mass rate of saltating particles per unit width, Q (kg/m/s), is assumed to be

deposited along a fetch of L meters long." This is correct and indeed how we conceptualize L.

To clarify the definition of *L*, we have added (see L156-157) the following brief explanation and reference to our Keenan et al. (2021) paper in which we go into greater detail on the saltation scheme.

"*L* can be conceptually understood to represent the distance over which the originally upwind and now saltating particles have been eroded from the snow surface (Keenan et al, 2021).".

l.124 (eq.4): The numerator of this equation corresponds to the expression proposed by Sørensen (1991) for the transport rate (see page 75 of the article, eq. 3.22). This expression is in units of g/cm/s (this is stated at the end of page 72 of the article, below equation 3.9). This poses a problem because the coefficients 0.0014 and 205 are not dimensionless values - 205 has velocity units (cm/s) and 0.0014 has units of s2/cm. If we want to express Q in units of kg/m/s, these factors should change to 2.05 and 0.14, respectively. The dimensionally correct expression predicts much higher values of Q and its validity to model snow saltation is still to be assessed. This issue with the Sørensen's expression was previously pointed out in the PhD thesis of Vionnet (2012), page 103, Fig. 5.3 (french only). In the mentioned PhD thesis as well as in Vionnet et al. (2014), the use of the latest expression of Sørensen (2004) is proposed. This can be a good option for Alpine3D as it does not deviate significantly from the dimensionally wrong Sørensen equation (see Fig.5.3 in the PhD thesis). Independently of the approach chosen by the authors, I believe it is advisable to present Q - the numerator of eq.4 - in a separate equation and cite the respective article.

Thank you for pointing this out. Indeed, after looking into the original reference, we agree that the magnitudes of the parameters in our implementation are incorrect. Furthermore, we have implemented your suggestion and now explicitly define the numerator of eq. 4 as Q.

Unfortunately this error seems to have been introduced in SNOWPACK by Lehning and Fierz (2008) and at this point remains unfixed. We have created an issue in our source code repository to be fixed in the future.
https://github.com/snowpack-model/snowpack/issues/24

Now, we would like to recall here that we build upon the work by Keenan et al. (2021) and Wever et al. (2022), who showed how the erosion and deposition calculated using the currently implemented code yielded satisfying results. The fact that an incorrect parameterization could yield satisfying results can be understood when considering that Q is already scaled by the poorly constrained tuning parameter L, such that any

changes in the formulation of Q could be accompanied by varying the fetch length L to maintain similarly good performance. Lastly, correcting this parameterization and updating the simulations for this manuscript would require substantial additional effort, that, in light of what we mention above, may have little impact on the conclusions in this work. Nevertheless, transparency of this issue is necessary, thus, in order to account for your astute (and very welcomed!) comment, we have added the following text (see L179-190):

"Additionally, it has come to our attention that SNOWPACK's parameterization of Q does not perfectly match the original parameterization proposed by Sørensen (1991). As noted in Vionnet (2012), the parameters 0.0014 and 205 in Eq. 4 reflect units for Q of g cm$^{-1}$ s$^{-1}$, whereas here we define Q with units of kg m$^{-1}$ s$^{-1}$. This implementation error in SNOWPACK was introduced by Lehning and Fierz (2008), but since we build upon the previous work by Keenan et al. (2021); Wever et al. (2022), who showed satisfying results for erosion and deposition calculated using the currently implemented code, we did not correct this error. Practically speaking, this error leads SNOWPACK to underestimate the magnitude of Q compared to the intended parameterization described in Sørensen (1991). However, it is also important to note here that Q is ultimately scaled by the poorly constrained tuning parameter L, such that any changes in the formulation of Q could require a new choice for the fetch length L to maintain similarly good performance. Thus, it is not certain that updating our parameterization of Q would lead to improved or more physically meaningful results. That said, future studies could consider using the updated parameterization of Q introduced in Sørensen (2004) which Vionnet (2012) showed to produce similar results to our presently dimensionally incorrect parameterization of Q (Vionnet et al., 2014)."

l.131-132 (point 1): The wind field at multiple vertical levels cannot be computed with WindNinja?

WindNinja could in fact save output at multiple vertical levels. However, what we try to illustrate here is that this is not necessary in our approach. We only need to save and calculate the wind speed at 10 m. This allows for a highly efficient coupling to the Alpine3D model. To make this more clear we have revised to the following (see L170-171):

"full prognostic solution of blowing snow transport via suspension requires numerically expensive calculations using wind vector fields at multiple vertical levels (Lehning et al., 2006, Sharma et al., 2021)"

l.133-134 (point 3): This is not advisable at high wind speeds because the aeolian transport of snow stops being governed by the wind field close to the ground alone. In addition, the saltaion velocity considered in eq.7 would have to be revised (it describes

saltation only as suspended particles are expected to have velocities comparable to the wind speed).

Given your remark and the remarks in the review by Charles Amory on the importance of suspension versus saltation, we now do agree that the argument that we use the fetch length to naively account for suspension is not as justified as we made it sound, as you pointed out. It is indeed better not to aim to include suspension in saltation given the different advection speeds. We rephrased this paragraph to better reflect these notions (see L173-178).

I.144 (eq.6): I believe the variable us should be defined in a more clear way: does it represent the particles speed or the wind speed in the saltation layer? Pomeroy and Gray (1990) proposed $2.8u*th$ as the average particle speed inside the saltation layer. However, if eq.6 is a mass conservation equation, where the quantity Ms is being advected by the flow, us should represent the wind speed. The wind speed in the saltation layer must be higher than the particles speed so that the particles are continuously accelerated. Do the authors consider these two quantities to be equal so that the mass in saltation is considered a passive scalar?

We indeed defined $u_s$ similar to Pomeroy and Gray (1990), so it represents the particle speed. To make this more clear, we now write (see L198): "In our implementation, $u_s$ represents the saltation particle speed and is defined as parallel to the 10m wind speed unit vector…" Since the wind speed concerns the speed of the molecules in the air, any quantity associated with that should indeed be advected with the wind speed (for example, temperature and humidity). However, the saltating particles do not travel with the same speed as the air molecules, but at a lower speed. This is due to the constant interaction of the particles with the surface, leading to decelerations and accelerations relative to the governing wind speed. So we cannot assume that the wind speed and particle speed are the same. So in this case, the quantity of interest is advected by the flow of saltating particles, not air particles, and we maintain that mass conservation is not violated in Eq. 6.

I.175-176: It is not clear if Alpine3D is ran for some time before the time period of interest in order to improve the initial state of the firn column. I suggest making it clear in the text.

Excellent point. Indeed we did not make this clear. Also the other review indicated that more explanation of the spinup procedure is required, which we added (see L240-L249). We do not run Alpine3D for any period of time before the analyzed time period (2015 - 2020). We have updated the text to the following in order to more clearly reflect this implementation (see L249-251):

"Alpine3D downscaling is then launched at the beginning of the analysis period (2015 in this study), meaning that although we initialize Alpine3D with a spun up firn column, its properties initially reflect the non-downscaled MERRA-2 climate."

l.191: Please specify over what years was the annual average performed.

Unfortunately, answering this question uniformly is not possible. In simple terms, these observations are produced by counting the number of annual layers in the top 50 m of the firn column. Because annual layer thickness varies spatially, the number of years over which the average is performed also varies spatially. We've added the following sentence (see L270-272):

"This product intends to represent the annual-average SMB. However, because of the finite thickness of firn isochrones and spatially variable accumulation rates, the annual average is calculated over varying periods depending on the location."

l.190-195: Can the authors say something about the uncertainty/accuracy of the snow accumulation predictions derived from the firn thickness?

The authors of the dataset do not provide quantitative uncertainty estimates for their annually averaged accumulation dataset. However, as stated in the text, the 25 km averages are set to that of MERRA-2. Thus, large scale uncertainties are controlled by MERRA-2.

That said, the authors do report relative accumulation errors that they describe as "negligibly small with a mean error of 0.002 (interpreted as 0.2%)." We have added the following to the text (see L272-273):

"No quantitative uncertainty estimates are assigned to the absolute SMB observations, however the authors report an average relative accumulation error of 0.2%."

Fig.4 and 5: The observations signal (red line) seems the same in Figures 4c and 5c. However, the simulations correspond to different time periods (2015 only vs 2015-2020). Is the observation signal indeed the same? If yes, to what time period does it correspond?

Great question. Yes, the observational transect is the same in Figures 4c and 5b and represents a long-term annual average (i.e., comprising recent decades because this is a relatively high accumulation area and the product is built using radar soundings of near-surface firn). The exact time frame this transect corresponds to is unknown because 1) it is not reported by Dattler et. al. 2019 and 2) uncertainty associated with firn isochrone tracking. We revised the discussion in Section 2.7, following previous comments (see above), which we hope would also make this more clear.

l.225: The authors assume R-squared to be a proxy of the variance explained. As this is not accurate for all distributions, I suggest the authors to justify this assumption.

Thank you for bringing this point to our attention. We now state in the manuscript that we assume that the modeled and observed SMB are linearly related. Since we use Ordinary Least Squares to perform the regression analysis to calculate $R^2$, we also can assume that the mean of the residuals is zero, such that $R^2$ corresponds to variance explained. We revised this in the manuscript (see L302-304).

l.245: Is the discrepancy between the Alpine3D and the MERRA-2 transect results completely explained by snow being drifted from the analysis domain to the 15km border? It is not clear why the comparison between Alpine3D and MERRA-2 along the whole analysis domain is directly related to the comparison between these two models along the transect.

You raise an excellent point. Thank you. Indeed the discrepancy between mean Alpine3D and MERRA-2 averaged over the analysis domain is largely explained by the divergence of drifting snow out of the analysis domain and into the 15 km border, which leaves less mass available for accumulation along the transect. To make this more clear, and to also point to the possibility of net divergence in the area due to the large scale wind field, we revised the sentence to (see L324-327):

"The discrepancy between MERRA-2 and Alpine3D along the transect is most likely explained by net divergence of saltating snow out of the analysis domain (Fig. 2), resulting in less mass available for accumulation along the transect. Since wind speeds generally increase from the top right to the bottom left over the model domain, some net divergence over the transect also may occur."

l.255-256: In the manuscript, surface mass balance variability is completely attributed to snow redistribution. Even though it is well known that snow redistribution plays an important role, it would be interesting to compare the outputs of the Alpine3D downscaling with and without the snow drift model. This would isolate the effect of snow redistribution from the effect of spatially varying heat fluxes.

Agreed. The other reviewer likewise raised this concern and we provide the same response here. In the revised manuscript we include an Alpine3D simulation for the year 2015 without horizontal snow redistribution (see newly added Fig. 10). In the revised manuscript, we discuss the effect of simulated snow redistribution on simulated SMB variability, over variability caused by other processes (see newly added Section 3.7, L397-405).

l.265: It would be interesting if the authors could present how the process of wind-driven compaction is modeled in Alpine3D. For example, are the properties of deposited snow prescribed? Or do they depend on the properties of the previously eroded snow?

This is an appropriate request and in line with a question from Reviewer 2. In order to answer your question as well as improve the readability of the manuscript, we have added the following text (see L164-168):

"Following erosion and subsequent redeposition, several snow microstructural properties are updated in SNOWPACK according to the "redeposit" scheme presented in Keenan et al., 2021. For example, the density of redeposited snow layers are parameterized according to wind speed (Eq. 4 in Keenan et al., 2021) while sphericity and dendricity are both set to 0.875. The grain radius and bond radius are set to 0.2 mm and 0.05 mm, respectively while albedo is defined by Eq. 7 in Groot Zwaaftink et al. (2013)."

Although certainly of some interest to this paper, because these developments have been published elsewhere, we believe that the full details of SNOWPACK's microstructural implementation are best left out of this manuscript. For this reason, we refer readers to Keenan et al. (2021) and Groot Zwaaftink et al. (2013) for further explanation.

l.317: The definition of SMB is not very clear in the Conclusions. The definition presented in the Introduction is more accurate. Please consider rephrasing.

Good suggestion. Notebally, this comment is consistent with the other reviewer's suggestion. We have revised to (see L408):

"The primary way ice sheets accumulate mass is through net snow accumulation at the ice sheet surface. SMB quantifies the balance between processes which accumulate and ablate mass at the surface of ice sheets."

**Technical corrections:**

l.44: The word "recent" is doubled.

Fixed. Thank you (see L63).

l.64-67: I suggest referring to the respective sections as in lines 60-64.

Good catch. Implemented (see L92-93).

l.69: I suggest revising the need for section 2.1. The content of this section is mainly an introductory paragraph of section 2. Hence, it can be included below the title "Methods" without the need for a new subsection.

Thank you for the suggestion. After careful consideration, we have decided to keep the subsection title as we believe it will be useful in guiding readers who are interested in finding a quick methodological summary.

Figure 1: It would be interesting to add the remaining applications to the scheme (e.g. MeteoIO, WindNinja, ICESat-2, MERRA-2).

Thank you for this suggestion, which we incorporated in the updated Figure 1.

l.91: Consider replacing "off" by "on".

Fixed. Thank you (see L121).

l.100: Consider replacing "cheaper" by "computationally lighter".

Good feedback, thank you. We have updated it to "computationally cheaper" (see L130).

l.120: Consider replacing "layers" by "snow layers".

Implemented. Thank you (see L151).

l. 135: Taking into account that Alpine3D is more than a wind redistribution module, this title might mislead the reader. Consider removing "Alpine3D:".

Agreed, even though we still think it is important to make clear at this point that the section describes the workings of Alpine3D. Therefore, we have rephrased the subsection header to: "Numerical treatment of deposition and erosion in Alpine3D" (see L191).

l.165: I suggest considering the option of moving subsections 2.5-2.7 to a new section. It can be called "Case Study", for example.

Thanks for bringing this idea to our attention. Although these three sections could certainly be considered descriptive of a single case study, we believe that they provide unique information that benefits from their granular heading. For this reason, we respectfully keep the 2.5 – 2.7 subsections.

l.184: Even though the meaning of "efficiency" is clear in the text, please keep in mind that it has a specific meaning in high performance computing (see parallel efficiency).

Good point. To avoid confusion, we have revised to "computational speed" (see L259).

l.184: I suggest writing all numbers in a consistent way: either delete the comma in number 27126 (l.182) or add it in number 1130 (1,130).

Good point. We now use commas in both cases (see L259).

l.185 and 187: Taking into account that Alpine3D includes SNOWPACK, I believe it is not vary precise to talk about "SNOWPACK and Alpine3D" in this context. Consider replacing by Alpine3D only.

Thanks. We have adopted this suggestion (see L259 and L261).

l.187: Considering replacing "cheaper". What about "computationally less expensive"?

We have adopted your suggestion and replaced "cheaper" with "computationally less expensive" (see L262). Thanks for pointing out this opportunity for improved clarity.

l.232: The word "decreasing" is misspelled.

Fixed. Thank you.

l.243: It is not very clear what is the "2015-2020 period" and the "long term average". Please consider rephrasing.

Good point, we are now more explicit after revising to (see L320-323):

"Furthermore, it is worth noting that over the length of the transect, MERRA-2 2015-2020 mean annual SMB exceeds that of observations (504 and 461mm w.e. yr$^{-1}$), indicating that the 2015-2020 simulated SMB exceeded the 1980-2017 MERRA-2 mean annual SMB (Section 2.7)."

l.268: Is it R or R-squared?

Indeed it is R. We report R instead of R-squared to make clear the positive correlation.

l.339: Consider adding "that" after "shown".

Added (see L430).

l.347-348: Did the authors consider adding the model to the gitlab of Alpine3D?

Thank you for the suggestion. Although certainly worth considering, we have decided not to because we have already made our version entirely open source (https://github.com/snowpack-model/snowpack/tree/driftingsnow) and we do not control the SLF gitlab source repository.

**Reference used here, but not cited in manuscript:**

Groot Zwaaftink, C. D., Mott, R., and Lehning, M. (2013b), Seasonal simulation of drifting snow sublimation in Alpine terrain, *Water Resour. Res.*, 49, 1581– 1590, doi:10.1002/wrcr.20137.

**A wind-driven snow redistribution module for Alpine3D v3.3.0: Adaptations designed for downscaling ice sheet surface mass balance**

Eric Keenan[1], Nander Wever[1,2], Jan T. M. Lenaerts[1], and Brooke Medley[3]

[1]Department of Atmospheric and Oceanic Sciences, University of Colorado, Boulder, CO, USA
[2]WSL Institute for Snow and Avalanche Research SLF, Davos, Switzerland
[3]Cryospheric Sciences Laboratory, NASA Goddard Space Flight Center, Greenbelt, MD, USA

**Correspondence:** Eric Keenan (eric.keenan@colorado.edu)

**Abstract.** Ice  sheet surface mass balance describes the net snow accumulation at the ice sheet surface. On the Antarctic ice sheet, winds redistribute snow resulting in surface mass balance that is variable in both space and time. Representing wind-driven snow redistribution processes in models is critical for local assessments of surface mass balance, repeat altimetry studies, and interpretation of ice core accumulation records. To this end, we have adapted Alpine3D, an existing distributed snow modeling framework, to downscale Antarctic surface mass balance to horizontal resolutions up to 1 km. In particular, we have introduced a new two-dimensional advection-based wind-driven snow redistribution module that is driven by an offline coupling between WindNinja, a wind downscaling model, and Alpine3D. We then show that large accumulation variability can be at least partially explained by terrain-induced wind speed variations which subsequently redistribute snow around rolling topography. By comparing Alpine3D to airborne-derived snow accumulation measurements within a testing domain over Pine Island Glacier in West Antarctica, we demonstrate that our Alpine3D downscaling approach improves surface mass balance estimates when compared to MERRA-2, a global atmospheric reanalysis which we use as atmospheric forcing. In particular, when compared to MERRA-2, Alpine3D reduces simulated surface mass balance root mean squared error by $23.4 \, \mathrm{mm \, w.e. yr^{-1}}$ (13%) and increases variance explained by 24%. Despite these improvements,

our results demonstrate that considerable uncertainty stems from the employed saltation model, confounding simulations of surface mass balance variability.

**1 Introduction**

Ice sheet surface mass balance (SMB) is the difference between mass accumulation and ablation processes at the surface of ice sheets (Lenaerts et al., 2019). Mass accumulation is composed of precipitation as well as condensation and deposition of atmospheric water vapor, whereas ablation processes remove mass from the ice sheet surface via meltwater runoff, both atmospheric and surface sublimation, and evaporation. Additionally, local SMB is influenced by  the process of wind-driven snow redistribution, which we refer to as deposition in the case of  local mass gain and erosion in the case of mass loss (Lenaerts and van den Broeke, 2012).

25  Wind-driven snow redistribution is generally divided into two categories, drifting snow, where snow particles are transported by the wind in the lowermost 2 m of the atmosphere, and blowing snow where snow is redistributed at heights greater than 2 m.

Drifting and blowing snow have been shown to have a substantial effect on Antarctic SMB spatial variability at scales ranging from tens of kilometers (Dattler et al., 2019; Kausch et al., 2020) to meters (Picard et al., 2019). In fact, both Dattler et al.

30  (2019) and Picard et al. (2019) have shown that local deposition can exceed annual precipitation. In addition to redistributing mass, drifting and blowing snow contribute to mass loss by promoting enhanced sublimation as snow particles are entrained in the lower atmosphere (Palm et al., 2017). In spite of drifting and blowing snow sublimation being a source of mass loss, evaluation of model simulation of these processes remains difficult (Amory et al., 2021). The net effect of drifting and blowing snow is preferential deposition in areas of mass convergence at the expense of areas with net divergence (Lehning et al., 2008).

35  Despite our lack of complete physical understanding of the processes which govern preferential deposition at small spatial and temporal scales (Comola et al., 2019b), deposition and erosion can be conceptually summarized as the local convergence of previously eroded snow particles from upwind minus locally eroded snow (Liston et al., 2007). Local erosion is governed by the direct competition between forces that act to erode snow and those which act to anchor snow to the ice sheet surface. Erosive forces, namely surface wind stress, are controlled by atmospheric boundary layer processes

40  (Paterna et al., 2016; Comola et al., 2019a), while cohesive forces are controlled by snow-microstructural properties including grain size and bond strength (Clifton et al., 2006; Melo et al., 2022). This interplay between boundary layer and snow-microstructural processes has historically motivated the development of tightly coupled atmospheric and land surface snow models (e.g., Lawrence et al., 2019; Amory et al., 2021; Sharma et al., 2023).

When the combined effect of surface wind stress and impact force from saltating particles exceeds cohesive forces at the

45  snow surface, saltation of snow particles is initiated or maintained within the lowermost 10 cm of the atmosphere  (Schmidt, 1980; Pomeroy and Gray, 1990). Within the saltation layer,  atmospheric momentum is entrained by snow particles which eventually collide with and subsequently mobilize additional particles at the snow surface. Above the saltation layer, a suspension layer can be present where turbulent eddies generate sufficient upward lift forces that can keep particles afloat against gravity (e.g., Bintanja, 2000). Measurements from Antarctica show that

50  particles in suspension are typically much smaller than those that are transported in saltation (Nishimura and Nemoto, 2005). Correspondingly, mass densities in the suspension layer are found to be orders of magnitude smaller than in the saltation layer. However, with increasing wind speed, the suspension layer can grow orders of magnitude larger than the height of the saltation layer, leading to estimates that the largest part of total mass transport during high wind speed events could be carried in the suspension layer (Pomeroy and Male, 1992; Liston and Sturm, 1998; Mann, 1998). It is difficult to quantify the contribution

55  of suspension to total mass transport on average, with some authors arguing that saltation accounts for 50 – 75% of total wind-driven mass transport (Gromke et al., 2014). However, although deposition and erosion are ubiquitous across much of the Antarctic ice sheet, it is possible that local SMB variability is primarily driven by a small number of high wind speed events which occur only a few times per year.

60 Nevertheless, the interaction between wind transport of snow and the snow surface  is governed by the saltation layer. As saltating particles break apart upon collision with the surface, snow grain fragmentation and rounding are observed, resulting in increased density  (Vionnet et al., 2012; Amory et al., 2021) in a processes referred to as drifting-snow compaction. Reliable model representation of drifting-snow compaction is now particularly attractive owing to recent  advancements in satellite altimetry technology (e.g. CryoSat-2 and ICESat-2). Vertical accuracy, spatial resolution, and ground

65 track repeat frequency have all increased, providing precise measurements of ice sheet surface height change (Smith et al., 2020). However, in order to reliably convert these height changes into mass, particularly over short time scales, we rely on accurate snow and firn density estimates from models (The IMBIE team, 2018). Thus, to confidently assign subtle observed changes in height to quantifiable changes in mass, our models must capture the complex spatial and temporal patterns of deposition and erosion.

70 Current state-of-the-art firn densification models, which are used to convert satellite observed volume changes into mass, successfully capture broad regional variability in firn properties, including density (Ligtenberg et al., 2011; Medley et al., 2022b). However, because of the relatively coarse horizontal resolutions at which they are applied ($5.5 - 35 \, \mathrm{km}$), these models are unable to represent spatial variability in firn processes, including deposition and erosion, at the horizontal scales now sampled by satellites ($< 1 \, \mathrm{km}$). This spatial gap between satellite observations and firn densification models may not be

75 immediately important for mass balance retrievals at continental scales (Verjans et al., 2021). Nevertheless, improved model representation of wind-driven snow redistribution at finer spatial scales can be used to constrain regional to local surface mass balance (Rignot et al., 2011), improve volume-to-mass conversions for repeat altimetry (Shepherd et al., 2012; Zwally et al., 2015), provide the ice coring and radar communities with a mechanism to select representative sampling locations for SMB reconstructions (Kausch et al., 2020), and inform future studies by providing baselines to estimate sublimation of drifting and

80 blowing snow.

To facilitate local SMB estimates and reconstructions as well as repeat altimetry interpretation, we present a new technique for dynamically downscaling Antarctic SMB by building upon the existing Alpine3D v3.3.0 model framework (Section 2.1). Alpine3D already contains a module for the 3D treatment of drifting and blowing snow (Lehning et al., 2006), but its use requires 3D wind fields at high temporal resolution from an external source, and it is computationally very demanding to

85 run. This motivates the current study to find a simpler, and therefore computationally lighter method to describe wind transport by snow, particularly since the fully 3D approach also does not guarantee good agreement between simulated and observed snow depth in severely wind affected areas (Mott et al., 2008, 2010; Gerber et al., 2017). To this end, we describe the use of WindNinja to downscale wind fields onto local topography (Section 2.2), and demonstrate the use of a one-dimensional snow model to diagnose local erosion (Section 2.3) and then distribute this mass horizontally across adjacent grid cells using a

90 new, two-dimensional advection-based redistribution module (Section 2.4). We then present downscaling results at 1 and 3 km horizontal resolution, and evaluate the added value by quantitatively comparing our results, along with a global SMB product, to a 130 km long airborne radar derived SMB transect in interior West Antarctica over Pine Island Glacier (Sections 3.1 - 3.2, 3.5).

**2 Methods**

 ### 2.1 Alpine3D: Surface mass balance downscaling framework

We use and further develop the existing Alpine3D v3.3.0 model framework (Lehning et al., 2006) to downscale Antarctic SMB processes to a target horizontal resolution of up to 1 km. At its core, the model framework exploits the MeteoIO library (Bavay and Egger, 2014) to handle meteorological preprocessing and downscaling (Section 2.2), SNOWPACK (Bartelt and Lehning, 2002; Lehning et al., 2002b, a), a physics-based land-surface snow model for the detailed description of snow microstructural properties (Section 2.3), and a new module to calculate horizontal mass fluxes between adjacent grid cells (sections 2.4 and 2.6, Fig. 1).

**2.2 Atmospheric forcing: Meteorological downscaling**

At the snow surface, we prescribe hourly MERRA-2 global atmospheric reanalysis (Gelaro et al., 2017) which we have down-scaled to the Alpine3D grid using the MeteoIO preprocessing and downscaling library (Bavay and Egger, 2014). These down-scaled time series include: 2 m air temperature, relative humidity, incoming shortwave and longwave radiation (ISWR and ILWR), precipitation rate, and 10 m wind speed and direction. 2 m air temperature is first downscaled to the Alpine3D grid using an ordinary kriging algorithm with a lapse rate of -6 °C km$^{-1}$ designed to capture a typical atmospheric lapse rate (Martin and Peel, 1978) while relative humidity is spatially interpolated according to Liston and Elder (2006). Precipitation rate and ISWR are then interpolated using inverse distance weighting, with ISWR undergoing a simple correction for slope and topographic shading (Helbig, 2009). ILWR is interpolated by first calculating the hourly average of all MERRA-2 grid cells within the domain, and then applying a constant lapse rate of -31.25 W m$^{-2}$ km$^{-1}$ . At the bottom of the firn column, we (Michel et al., 2022). We follow Keenan et al. (2021), by applying the MERRA-2 mean annual surface temperature as a Dirichlet thermodynamic boundary condition at the bottom of the firn column. This assumption is supported by observations from the dry snow zone of the Greenland ice sheet, where differences between mean annual air temperature and firn temperature at 10m were found to be typically within a few tenths of a degree (Alley and Koci, 1990). Ligtenberg (2014) also shows in a model result that the seasonal cycle in firn temperature disappears around 10m depth. Note that for all topographic calculations, we use an ICESat-2 derived digital elevation model (DEM) (Medley et al., 2022a).

Because reliable simulations of deposition and erosion require accurate wind speed and direction fields (Reynolds et al., 2021), we test two approaches for downscaling wind speed and direction from the relatively coarse (0.5° latitude × 0.625° longitude) MERRA-2 grid. First, we apply the terrain-based index method proposed by Liston et al. (2007) which adjusts wind speed and direction based off on topographic exposure and sheltering. Note that we use the default slope and curvature weighting factors and choose a topographic length scale of 5 km. However, because Reynolds et al. (2021) showed that simulated snow depth better captured observations when forced with WindNinja downscaled wind fields (Forthofer et al., 2014) compared to the relatively simpler terrain-based index methods presented in Liston et al. (2007), we have implemented WindNinja (Version 3.7.1) as an alternative offline wind speed and direction downscaling technique within the Alpine3D modeling framework. WindNinja is a finite element diagnostic model which leverages a mass-conservation solver and DEM

[Figure]

**Figure 1.** Anatomy of an Alpine3D time step: Hourly global atmospheric reanalyis (a) is combined with ICESat-2 derived surface topography (b) to calculate downscaled meteorology (c, d) at each SNOWPACK model grid cell. The SNOWPACK model is then used to calculate snow microstructural properties at each grid cell (d̶e̲). Finally, the Alpine3D model is used to calculate horizontal mass transport across adjacent grid cells (e̶f̲) which is then sent back to SNOWPACK for the next time step. Note that in (c,d), downscaled wind speed and 2 m air temperature are shown as an example, while also relative humidity, incoming shortwave and longwave radiation, precipitation rate, and 10 m wind direction are downscaled (see text).

to simulate the mechanical effects of terrain on the flow. In terms of WindNinja model configuration, we have selected the "fine" finite element mesh resolution and chosen "grass" as our surface roughness category (snow is not currently an option). Despite representing a significant increase in complexity compared to other terrain-based interpolation techniques (e.g., Liston 130  et al., 2007), WindNinja is still s̶i̶g̶n̶i̶f̶i̶c̶a̶n̶t̶l̶y̶ computationally cheaper than high-resolution numerical weather models (e.g.

the Weather Research and Forecasting (WRF) Model), which solve the non-hydrostatic, fully compressible Navier-Stokes equations at multiple vertical levels (Wagenbrenner et al., 2016), and can therefore be run with reasonable computational resources (Section 2.6).

**2.3 SNOWPACK: One-dimensional physics-based snow model**

135 We use SNOWPACK, a physics-based land-surface snow model, to describe one-dimensional snow and firn processes at each Alpine3D grid cell. SNOWPACK was originally developed for avalanche warning applications, and has been continuously enhanced in order to represent various cryospheric processes including seasonal snow (Sharma et al., 2023), sea ice snow cover (Wever et al., 2020), and polar firn compaction (Groot Zwaaftink et al., 2013; Keenan et al., 2021). In this study, we use the SNOWPACK physics presented in Keenan et al. (2021), with one exception being a new parameterization for surface

140 roughness length, $z_0$ (m), Eq. (1), tuned to observed seasonal variability between winter and summer surface roughness in coastal East Antarctica (Amory et al., 2017). In Eq. (1), $\kappa$ is the von Kármán constant (0.4), $T_{2\,m}$ is the 2 m air temperature (°C), and $C_{\mathrm{DN10}}$ is the neutral drag coefficient at 10 m, Eq. (2).

$$
z_0 = \begin{cases} \dfrac{10}{\exp \frac{\kappa}{\sqrt{C_{\mathrm{DN10}}}}} & T_{2\mathrm{m}} > -20\,°C \\ 0.0002 & T_{2\mathrm{m}} \leq -20\,°C \end{cases} \tag{1}
$$

$$
C_{\mathrm{DN10}} = 2.7 \times 10^{-3} + 9.0 \times 10^{-5} T_{2\mathrm{m}} + 1.5 \times 10^{-6} T_{2\mathrm{m}}^2 \tag{2}
$$

145 In SNOWPACK, snow redistribution is initiated when the friction velocity $u_*$ (m s$^{-1}$) exceeds the surface threshold friction velocity $u_{*\mathrm{th}}$ (m s$^{-1}$, Eq. (3)), which is calculated diagnostically as a function of snow microstructural properties including snow grain sphericity $SP$ (0 - 1), radius $r_\mathrm{g}$ (m), bond radius $r_\mathrm{b}$ (m), and coordination number $N_3$ (Lehning and Fierz, 2008).

$$
u_{*\mathrm{th}} = \sqrt{\frac{A\rho_\mathrm{i} g r_\mathrm{g}(SP+1) + B\sigma N_3 \frac{r_\mathrm{b}^2}{r_\mathrm{g}^2}}{\rho_\mathrm{a}}} \tag{3}
$$

In Eq. (3), $\rho_\mathrm{i}$ is the density of ice (917 kg m$^{-3}$), $\rho_\mathrm{a}$ is the density of air (1.1 kg m$^{-3}$), $g$ is the gravitational acceleration
150 (9.8 m s$^{-2}$), $\sigma$ is a reference shear strength set to 300 $Pa$, while constants $A$ and $B$ are set to 0.02 and 0.0015 respectively. Once drifting snow is initiated, snow layers are eroded from the top of the simulated snow cover until the saltation mass flux Φ (transport rate $Q$ (kg m$^{-1}$ s$^{-1}$), Eq. (4)) is satisfied . These eroded layers are then made available to Alpine3D for horizontal redistribution across adjacent grid cells (Section 2.4). In Eq. (4) , L is , is satisfied (Sørensen, 1991). The required local erosion is calculated from $Q$ by scaling to an erosion mass flux Φ (kg m$^{-2}$ s$^{-1}$) by dividing $Q$ by a characteristic horizontal length

155 scale $L$, which we call the fetch length, over which the saltating particles in Φ $Q$ are scaled to calculate local erosion Φ from the firn layer. $L$ can be conceptually understood to represent the distance over which the originally upwind and now saltating

particles have been eroded from the snow surface (Keenan et al., 2021). These eroded snow layers are then made available to Alpine3D for horizontal redistribution across adjacent grid cells (Section 2.4).

$$\Phi = \frac{Q}{L} = \frac{0.0014\rho_\mathrm{a}u_*(u_* - u_{*\mathrm{th}})(u_* + 7.6u_{*\mathrm{th}} + 205)}{L} \tag{4}$$

160    In contrast to our previous study (Keenan et al., 2021), we revert $L$ from 10 m back to the original SNOWPACK value of 70 m. We make this choice because Alpine3D simulations with a fetch length of 30 m or less were found to significantly overestimate the magnitude of deposition and erosion when compared to observations (Section 3.6). However, because we are not aware of any direct observations, $L$ can be effectively considered a tuning parameter whose magnitude is inversely proportional to the amount of eroded mass accounted for in the saltation mass flux $\Phi$ (Wever et al., 2022). Following erosion

165    and subsequent redeposition, several snow microstructural properties are updated in SNOWPACK according to the "redeposit" scheme presented in (Keenan et al., 2021). For example, the density of redeposited snow layers are parameterized according to wind speed (Eq. 4 in Keenan et al. (2021)) while sphericity and dendricity are both set to 0.875. The grain radius and bond radius are set to 0.2 mm and 0.05 mm, respectively while albedo is defined by Eq. 7 in Groot Zwaaftink et al. (2013).

    Note that we do not explicitly account for blowing snow suspension in our model. We make this pragmatic decision pri-

170    marily  because full prognostic solution of blowing snow transport via suspension requires numerically expensive calculations using wind vector fields at multiple vertical levels  (e.g., Lehning et al., 2006; Sh , while we discussed before that few observational case studies would be available for validation that quantitatively partition between saltation and suspension-driven transport and their eventual effect on deposition and erosion. It is also important to consider that the control provided by the poorly constrained fetch length $L$

175     would make it possible to locally erode more snow, which naively could be thought of as considering a suspension layer (Keenan et al., 2021), even though the advection distance of snow in suspension is larger given the higher advection speeds and longer airborne times of suspended particles.

    Additionally, it has come to our attention that SNOWPACK's parameterization of $Q$ does not perfectly match the original

180    parameterization proposed by Sørensen (1991). As noted in Vionnet (2012), the parameters 0.0014 and 205 in Eq. 4 reflect units for $Q$ of $\mathrm{g\,cm^{-1}\,s^{-1}}$, whereas here we define $Q$ with units of $\mathrm{kg\,m^{-1}\,s^{-1}}$. This implementation error in SNOWPACK was introduced by Lehning and Fierz (2008), but since we build upon the previous work by Keenan et al. (2021); Wever et al. (2022) , who showed satisfying results for erosion and deposition calculated using the currently implemented code, we did not correct this error. Practically speaking, this error leads SNOWPACK to underestimate the magnitude of Q compared to the intended

185    parameterization described in Sørensen (1991). However, it is also important to note here that Q is ultimately scaled by the poorly constrained tuning parameter L, such that any changes in the formulation of Q could require a new choice for the fetch length L to maintain similarly good performance. Thus, it is not certain that updating our parameterization of Q would lead to improved or more physically meaningful results. That said, future studies could consider using the updated parameterization

of $Q$ introduced in Sørensen (2004) which Vionnet (2012) showed to produce similar results to our presently dimensionally
incorrect parameterization of $Q$ (Vionnet et al., 2014).

**2.4   Numerical treatment of deposition and erosion in Alpine3D**

At each time step, the saltation mass flux $\Phi$ (see Eq. (4)) at each grid cell is scaled to a saltation mass $M_{\mathrm{s}}$ ($\mathrm{kg\,m^{-2}}$) by multiplying $\Phi$ by the Alpine3D time step $\Delta t_{\mathrm{A3D}}$ (s) (Eq. (5)).

$$M_{\mathrm{s}} = \Phi \Delta t_{\mathrm{A3D}} \tag{5}$$

$M_{\mathrm{s}}$ is then made available to Alpine3D for downstream redistribution , resulting in a local saltation mass perturbation $\Delta M_{\mathrm{s}}$ ($\mathrm{kg\,m^{-2}}$) which is positive in the case of net deposition and negative in the case of erosion. We calculate $\Delta M_{\mathrm{s}}$ by treating wind-driven snow redistribution as a two-dimensional horizontal advection problem (Eq. (6)) where $\boldsymbol{u_{\mathrm{s}}}$ is the saltation velocity vector ($\mathrm{m\,s^{-1}}$). In our implementation, $\boldsymbol{u_{\mathrm{s}}}$ represents the saltation particle speed and is defined as parallel to the 10 m wind speed unit vector $\hat{\boldsymbol{U}}_{10\,\mathrm{m}}$ with a magnitude parameterized as a function of $u_{*\mathrm{th}}$ (Eq. (7)) according to Pomeroy and Gray (1990).

$$\frac{\partial M_{\mathrm{s}}}{\partial t} + \boldsymbol{u_{\mathrm{s}}} \cdot \nabla M_{\mathrm{s}} = 0, \tag{6}$$

$$\boldsymbol{u_{\mathrm{s}}} = 2.8 u_{*\mathrm{th}} \hat{\boldsymbol{U}}_{10\,\mathrm{m}} \tag{7}$$

We solve for $\Delta M_{\mathrm{s}}$ by numerically integrating Eq. (6) forward in time using a first-order accurate, upwind finite difference scheme with an adaptive sub-time step $\Delta t_{\mathrm{CFL}}$ (s) in order to ensure numerical stability under the Courant-Friedrichs-Lewy (CFL) condition (Courant et al., 1928). The local saltation mass perturbation at the $i^{\mathrm{th}}$ sub-time step originating from advection of saltating snow in the $\hat{x}$ and $\hat{y}$ directions $\delta M_{\mathrm{s},x}^{i}$, $\delta M_{\mathrm{s},y}^{i}$ ($\mathrm{kg\,m^{-2}}$, Eqs. (8) and (9)), are then calculated with $\Delta X$ being the Alpine3D horizontal grid resolution.

$$\delta M_{\mathrm{s},x}^{i} = \begin{cases} \frac{\left(M_{\mathrm{s}(x-1,y)} - M_{\mathrm{s}(x,y)}\right)}{\Delta X} (\boldsymbol{u_{\mathrm{s}}} \cdot \hat{x}) \Delta t_{\mathrm{CFL}} & (\boldsymbol{u_{\mathrm{s}}} \cdot \hat{x}) > 0 \\ \frac{\left(M_{\mathrm{s}(x,y)} - M_{\mathrm{s}(x+1,y)}\right)}{\Delta X} (\boldsymbol{u_{\mathrm{s}}} \cdot \hat{x}) \Delta t_{\mathrm{CFL}} & (\boldsymbol{u_{\mathrm{s}}} \cdot \hat{x}) < 0 \end{cases} \tag{8}$$

$$\delta M_{\mathrm{s},y}^{i} = \begin{cases} \frac{\left(M_{\mathrm{s}(x,y-1)} - M_{\mathrm{s}(x,y)}\right)}{\Delta X} (\boldsymbol{u_{\mathrm{s}}} \cdot \hat{y}) \Delta t_{\mathrm{CFL}} & (\boldsymbol{u_{\mathrm{s}}} \cdot \hat{y}) > 0 \\ \frac{\left(M_{\mathrm{s}(x,y)} - M_{\mathrm{s}(x,y+1)}\right)}{\Delta X} (\boldsymbol{u_{\mathrm{s}}} \cdot \hat{y}) \Delta t_{\mathrm{CFL}} & (\boldsymbol{u_{\mathrm{s}}} \cdot \hat{y}) < 0 \end{cases} \tag{9}$$

210    $\Delta t_{\text{CFL}}$ is optimized such that we choose the maximum possible sub-time step while simultaneously satisfying the CFL condition (Eq. (10)).

$$(|\, \boldsymbol{u_s} \cdot \hat{x}\,| + |\, \boldsymbol{u_s} \cdot \hat{y}\,|)\frac{\Delta t_{\text{CFL}}}{\Delta X} < 1 \tag{10}$$

Note that in the case of the final sub-time step, we reduce $\Delta t_{\text{CFL}}$ such that the sum of all $N$ sub-time steps is equal to $\Delta t_{\text{A3D}}$. By summing equations 8 and 9 across all $\Delta t_{\text{CFL}}$ within $\Delta t_{\text{A3D}}$, we are able to finally calculate the local saltation mass

215    perturbation $\Delta M_{\text{s}}$ ($\text{kg}\,\text{m}^{-2}$, Eq. (11)) which is ultimately sent to SNOWPACK as a mass flux available at the next time step.

$$\Delta M_{\text{s}} = \sum_{i=1}^{N}(\delta M_{\text{s},x}^{i} + \delta M_{\text{s},y}^{i}) \tag{11}$$

Our upwind finite difference scheme cannot be applied at the edges of a domain, therefore we must prescribe boundary conditions. In this application (Section 2.5) we simulate a finite spatial domain. In order to force our domain to behave more like an infinite domain, we implement periodic boundary conditions (Hames et al., 2022) which implies that deposition and

220    erosion integrate to zero throughout the modeling domain. In areas where a net blowing snow mass influx or outflux with respect to the model domain may occur, for example in coastal regions where mass can be lost to the ocean, as well as in areas with steep topography, this assumption is not expected to hold. However, note that establishing boundary conditions for a blowing snow model in such areas would be very challenging. As discussed before, we may underestimate the effects of snow transport by wind by neglecting the dynamics of the suspension layer, including vertical advection of particles between

225    the saltation and suspension layer, which may impact spatial variability in SMB. Similarly, small scale atmospheric processes, like orographic lifting at scales not represented in the forcing data (Lenaerts et al., 2014), as well as complex flow patterns that can occur in steep terrain (Gerber et al., 2017), may create larger variability in SMB than we can simulate with our approach. Furthermore, we note that our model currently makes a significant simplification by not considering sublimation of snow particles actively entrained in the atmosphere above the snow surface (Amory et al., 2021).

230    ## 2.5 Pine Island Glacier experimental domain

As a demonstration of our Alpine3D downscaling framework, we define a 198 km × 137 km modeling domain centered over the Pine Island Glacier basin in West Antarctica (Fig. 2a, red rectangle). In this domain, surface elevations range from approximately 700–1500 m leading to relatively large surface slopes (mean: 0.64°, 10% quantile: 0.18°, 90% quantile: 1.3°) typical of the Antarctic escarpment zone. Strong and steady winds driven by consistent katabatic forcing, combined with rolling

235    topography, have been shown to drive SMB variability of up to a factor of five (200–1000 $\text{mm}\,\text{w.e.yr}^{-1}$) over horizontal scales on the order of 10 km (Dattler et al., 2019). In fact, Dattler et al. (2019) found the largest SMB spatial variability among their 17,500 km of flight line measurements in this domain (Section 2.7). This region therefore provides an opportunity to evaluate our SMB downscaling framework, while still being of modest spatial extent and therefore acceptable computational cost necessary for continuous model development. In terms of model initialization, we borrow from

240   We initialize the model based on the approach employed in our previous study  (Keenan et al., 2021), where good performance for density and temperature was obtained. For each MERRA-2 grid point (Section 2.2) inside the domain, we repeatedly ran a one-dimensional SNOWPACK simulation (Section 2.3)  for the period 1980–2017, while for every iteration taking the simulated firn layer by the end of 2017 as initial conditions

245   at the start of 1980. Those spin-up simulations describe the impact of drifting snow on the surface density and microstructure (Keenan et al., 2021; Wever et al., 2022), but they do not consider spatial snow redistribution. The process was repeated until the firn layer was at least 100 m deep by the end of 2017. We then ran one final simulation from January 1, 1980 until January 1, 2015. Afterward, each Alpine3D model grid point was initialized using the firn properties from the closest MERRA-2 grid point SNOWPACK simulation. Alpine3D downscaling is then launched at the beginning of the analysis period

250   (2015 in this study), meaning that although we initialize Alpine3D with a spun up firn column, its properties initially reflect the non-downscaled MERRA-2 climate. Note that in order to minimize the relative importance of boundary conditions for drifting snow, upon analysis we remove all grid cells within 15 km of the four domain boundaries, resulting in grid size of 168 km × 107 km (Fig. 2b black dashed rectangle).

**2.6   Computational parallelization and benchmarking**

255   To enable efficient numerical parallelization, the Alpine3D modeling framework supports a hybrid OpenMP/MPI implementation. For benchmarking, we perform calculations on a 1 km horizontal resolution 27,126 km$^2$ domain (Section 2.5) using general purpose compute nodes with 24 parallel CPUs per node. In the case of offline WindNinja wind speed and direction downscaling, we use 1 node (OpenMP, 24 CPUs) for 1 hour per calendar year, resulting in a computational speed of 1,130 km$^2$ yr (CPU hr)$^{-1}$. Whereas for  Alpine3D caclulations, we leverage 4

260   nodes (hybrid OpenMP/MPI, 96 total CPUs) for 18 hours per calendar year, netting  a computational speed of 16 km$^2$ yr (CPU hr)$^{-1}$. WindNinja is therefore at least one order of magnitude  computationally less expensive than Alpine3D, meaning that its added complexity is justified if downscaled SMB estimates are significantly improved.

**2.7   Surface mass balance observations**

265   To evaluate our SMB downscaling, we compare with a 130 km long, radar-derived accumulation transect (Fig. 2b black line T–T') developed by Dattler et al. (2019). This annual-averaged snow accumulation product captures spatial variability in along-track accumulation by tracking fluctuations in radar-derived firn isochrone thickness. Horizontal variations in thickness are then converted to mass using the Herron and Langway (1980) firn density model. At 25 km horizontal resolution, Dattler et al. (2019) set the mean annual (1980–2017) accumulation to that of MERRA-2 atmospheric reanalysis. While at smaller scales, accumu-

270   lation is modified by the combination of firn isochrone thickness and simulated firn density. This product intends to represent the annual-average SMB. However, because of the finite thickness of firn isochrones and spatially variable accumulation rates,

[Figure]

**Figure 2.** Alpine3D modeling and analysis domains in the EPSG 3031 coordinate system: (a) The 198 km × 137 km Alpine3D modeling domain (red rectangle) is centered over the Pine Island Glacier basin in West Antarctica. (b) Surface topography of Alpine3D modeling (red) and analysis (black dashed) domains with the location of the 130 km observed SMB transect (black line, T–T'). Panel (a) was created using the Norwegian Polar Institute's Quantarctica package (Matsuoka et al., 2018).

the annual average is calculated over varying periods depending on the location. No quantitative uncertainty estimates are assigned to the absolute SMB observations, however the authors report an average relative accumulation error of 0.02%.

**3  Results and Discussion**

**3.1  Impact of wind downscaling method on simulated surface mass balance**

To demonstrate the differences between downscaling techniques, we have downscaled MERRA-2 winds from the year 2015 using both the Liston and WindNinja (Section 2.2) algorithms to the 1 km Alpine3D grid (Fig. 3). WindNinja predicts a higher average wind speed than Liston (6.5 m s$^{-1}$ vs. 6.1 m s$^{-1}$) and is more consistent with MERRA-2 (6.6 m s$^{-1}$), which

[revised manuscript text omitted]

Finally, we performed a simulation without activating the newly developed module for wind redistribution of snow (Fig. 10), while keeping the WindNinja downscaling for wind as well as the downscaling approaches for the other variables. We find that in this case, the SMB field (Fig. 10a) is strongly dominated by the large scale gradient in SMB resulting from the MERRA-2

400

[Figure]

**Figure 9.** Connection between prescribed fetch length and simulated SMB. 2015 simulated SMB with (a) 30, (b) 70, and (c) 110 m fetch length. (d) Transect of observed (red), and simulated SMB with 30 (black), 70 (blue), and 110 m (magenta) fetch length.

forcing data, while lacking the small scale spatial variability found in the simulation with drifting snow shown in Fig. 4b. It illustrates that the Alpine3D simulation does not create any substantial small scale variability in SMB that could arise

[Figure]

**Figure 10.** Comparing the simulation with and without wind redistribution of snow for the year 2015. (a) 2015 Alpine3D SMB without wind redistribution of snow, (b) Transect of observed (red), 2015 Alpine3D with wind redistribution (blue), 2015 Alpine3D without wind redistribution (black) SMB. (c, d) Scatter plot of (c) 2015 with wind redistribution and (d) 2015 without wind redistribution vs. observations.

from other processes. For example, wind speed also controls surface sublimation, but the effect on SMB is not noticeable on larger scales. We conclude that the simulations including the wind redistribution of snow attribute virtually all of the spatial heterogeneity in SMB to snow redistribution processes by the wind (Fig. 4c, d).

**4  Conclusions**

The primary way ice sheets  accumulate mass is through net snow accumulation at the ice sheet surface. SMB quantifies the balance between processes which accumulate and ablate mass at the surface of ice sheets. Throughout the last two decades, regional climate models have been progressively developed to simulate large-scale ice sheet SMB fields.

However, recent advancements in observational capabilities (e.g. repeat satellite altimetry and airborne snow radar) indicate that SMB varies predictably in the presence of rolling topography. Despite the importance of capturing this SMB variability for assigning a mass change to repeat altimetry measurements, regional climate models are not capable of resolving this variability. To begin to close the gap between observations and existing regional climate models, and therefore improve local SMB estimates and support interpretation of repeat altimetry, we have expanded upon the Alpine3D framework in order to dynamically downscale Antarctic SMB.

In our implementation of Alpine3D, we leverage MeteoIO and WindNinja software to downscale meteorology onto the native Alpine3D grid. At each grid cell, the SNOWPACK snow model is used to diagnose snow properties including, density, temperature, grain size, and other snow microstructure parameters. With this information, SNOWPACK calculates a saltation mass flux for each time step and then sends this information to Alpine3D, where horizontal snow redistribution is treated as a two-dimensional advection problem. WindNinja downscaling results suggest that rolling topography with surface slopes between 0.25 - 2.0° are responsible for subtle terrain induced mechanical effects that vary mean wind speeds by $0.1 \ \mathrm{m\,s^{-1}}$. These wind speed perturbations ultimately drive spatially variable patters of deposition and erosion whose mean magnitude corresponds to 23% of mean annual accumulation.

By comparing 2015 – 2020 mean annual SMB downscaling results over a portion of Pine Island Glacier, West Antarctica to a 130 $\mathrm{km}$ long observational SMB transect, we found that, when compared to MERRA-2, Alpine3D reduces typical SMB errors by 23.4 $\mathrm{mm\,w.e.}$ (13%) and increases variance explained by 24%. Furthermore, we show that Alpine3D produces a similarly skillful downscaled SMB product when run at 3 $\mathrm{km}$ horizontal resolution. This suggests that Alpine3D can be run at a coarser resolution than what is primarily shown in this study (1 $\mathrm{km}$ horizontal resolution), therefore considerably reducing computational cost. Finally, despite the promising downscaling results presented in this study, Alpine3D still relies on the fetch length tuning parameter to scale the saltation mass flux. We have shown that the magnitude of deposition and erosion, and consequently SMB, are sensitive to the choice of fetch length. Because it is at the very least difficult, and quite likely impossible, to uniformly optimize the fetch length, we suggest that future studies migrate towards diagnosing the saltation mass flux purely from known quantities including snow properties and surface wind stress. Further, we note that our implementation makes important simplifications by neglecting drifting and blowing snow sublimation as well as horizontal redistribution by way of suspension. In regions where these processes are significant drivers of local SMB, future downscaling efforts would likely be improved by inclusion of these processes into Alpine3D.

*Code and data availability.* MeteoIO, SNOWPACK and Alpine3D are software published under the GNU LGPLv3 license by the WSL Institute for Snow and Avalanche Research SLF, Davos, Switzerland at https://gitlabext.wsl.ch/snow-models. The repository used to develop the versions of MeteoIO, SNOWPACK and Alpine3D used in this study can be accessed at https://github.com/snowpack-model/snowpack with the exact version corresponding to commit f023b9f archived at https://doi.org/10.5281/zenodo.5914787. MERRA-2 atmospheric reanalysis is available at https://gmao.gsfc.nasa.gov/reanalysis/MERRA-2/ and can be retrieved and processed using our workflow available at https://github.com/EricKeenan/download_MERRA2 and archived at https://doi.org/10.5281/zenodo.4560825. WindNinja source code is hosted at

https://github.com/firelab/windninja while the version used in this study is permanently archived at https://doi.org/10.5281/zenodo.4474633. Software used to produce the numerical simulations, analyses, and figures presented in this study are archived at https://doi.org/10.5281/
445 zenodo.5914751 and https://doi.org/10.5281/zenodo.5914727. Airborne snow accumulation radar data used in this study are archived at https://doi.org/10.5281/zenodo.3534315.

*Author contributions.* All authors contributed to experimental design, analysis, and manuscript preparation. Eric Keenan carried out Alpine3D simulations, performed the analysis, and produced the figures. Eric Keenan and Nander Wever both contributed to Alpine3D model development.

450 *Competing interests.* The authors declare that they have no conflict of interest.

*Acknowledgements.* The authors thank Charles Amory and an anonymous reviewer for their thoughtful comments which helped to improve the manuscript considerably. Eric Keenan, Nander Wever, Jan T. M. Lenaerts, and Brooke Medley acknowledge support from the National Aeronautics and Space Administration (NASA), grant 80NSSC18K0201 (ROSES-2016: studies with ICESat-2 and CryoSat-2). This work utilized the RMACC Summit supercomputer, which is supported by the National Science Foundation (awards ACI-1532235 and ACI-
455 1532236), the University of Colorado Boulder, and Colorado State University. The Summit supercomputer is a joint effort of the University of Colorado Boulder and Colorado State University. Data storage supported by the University of Colorado Boulder "PetaLibrary".

---

## Author Comment (AC2)

Thank you to editor Fabien Maussion as well as Charles Amory and the other, anonymous, reviewer for donating their time and energy to our manuscript. In order to address all comments, we have responded to each of them individually in blue. Line numbers in our responses refer to the track-changed version of the revised manuscript. Some newly added citations in our responses can be found in the revised manuscript.

We apologize for the very long delay in responding to the reviewers comments and revising the manuscript. We thank the editor, reviewers and editorial office for their patience and understanding that this is the unfortunate result of the first, second and last author having transitioned between jobs, as well as moving, in the last year.

On behalf all authors,

Eric Keenan

**Review #2 - Charles Amory**

**General comments**

This paper describes a new computational chain for downscaling the surface mass balance (SMB) over ice sheets, by combining the snowpack model SNOWPACK driven by MERRA2 reanalysis with an offline coupling between the wind downscaling model WindNinja and the wind redistribution model Alpine3D. This numerical design, while maintaining attractive computational costs, provides promising results by partially resolving the spatial variability in SMB due to wind redistribution of snow along a 130-km long transect in West Antarctica along which radar-derived SMB retrievals are available. Sensitivity to horizontal resolution and to a prescribed, yet influential parameter not constrained by observations, is explored. The proposed method is innovative and clearly fits with the scope of GMD, some limitations are addressed and the work is put in perspective of future developments. Furthermore, the study it concise and very pleasant to read. I think the paper deserves publication after the authors have addressed the following comments. I would particularly advise further discussion of the limitations of the modelling approach, notably regarding lacking suspension, atmospheric sublimation, feedbacks with the atmosphere and the attribution of the spatial variability in SMB to mostly saltation transport, in order to reach the level of rigour of the other discussion elements of the manuscript.

We thank Charles Amory for the positive feedback, and all the comments and suggestions. We improved the discussion on the limitations of our modeling approach, and also included some broader context of the results. Please find our detailed response to the issues raised below.

**Specific comments**

Abstract, L1: Snow accumulation is more the resultant of the SMB than a component of the SMB. The actual main components that lead to snow accumulation are snowfall, condensation/deposition and wind-driven snow deposition. Would you please reformulate a little to get this clearer?

Thank you for pointing this out. Indeed we agree that our formulation was rather imprecise. For improved clarity and readability, we have revised the first sentence to (see L1):

"Ice sheet surface mass balance describes the net snow accumulation at the ice sheet surface."

L24-26: You should mention atmospheric sublimation as a source of surface mass loss when defining the SMB. Currently the definition in the first paragraph only describes wind redistribution.

We intended sublimation to be inclusive of both atmospheric and surface sublimation. But in retrospect it makes sense to make this explicit. We have revised the mentioned sentence in the first paragraph of the introduction to (see L20-24):

"Mass accumulation is composed of precipitation as well as condensation and deposition of atmospheric water vapor, whereas ablation processes remove mass from the ice sheet surface via meltwater runoff, both atmospheric and surface sublimation, and evaporation. Additionally, local SMB is influenced by wind-driven snow redistribution, which we refer to as deposition in the case of local mass gain and erosion in the case of mass loss (Lenaerts and van den Broeke, 2012)."

L39-40: Would this assertion still be valid knowing that blowing snow layers can extend up to hundreds of meters above ground in Antarctica (Palm et al., 2017)? I find this assertion actually quite questionable, and one can expect the ratio of saltating to suspend snow mass to depend a lot on the area considered, and then to be very specific to the local snowpack, topographic and atmospheric conditions, or to the boundary conditions of the experiments. I did not find the materials in the referenced literature necessary to support such a statement. Gromke et al. (2014)'s results, which are based on wind tunnel experiments of limited dimensions, would not account for the well-developed bowing snow layers of hundreds of meters in depth, in which transport in suspension most likely dominate over saltation. Beside, these numbers are actually not demonstrated by Gromke et al. (2014), but just mentioned in the introduction and borrowed from Kind (1990), with no reason to consider it universally valid, especially when translated into an Antarctic environment. Moreover, contrary statements can be found in the literature. See for instance Bintanja (2000) p345: "Most of the snow transport occurs when snow is in suspension, with the saltation transport becoming rapidly less important as wind speeds increase (Pomeroy and Male, 1992; Mann, 1998).". Please nuance and adapt your sentence accordingly.

We agree that with the reviewer's remarks in mind, our statement was not sufficiently nuanced. There is indeed research indicating that at high wind speeds, suspension seems to carry most of the mass. We now include those studies in the revised manuscript (see L54). We also note at this point that for example Mann et al. (1999) indeed concludes that: "Blowing snow mass transport whilst in suspension was shown to dominate over transport by saltation.". However, this statement was based on measurements in the suspension layer while applying the Pomeroy (1988) model for the saltation layer, while Melo et al. (2021) suggest that the Pomeroy model may

underestimate mass transport in saltation. Thus, we feel that we can not confidently claim that also long-term average transport is dominated by suspension. Nishimura and Nemoto (2005) show measurements from Antarctica and calculated that the mass transport expressed in volume in the suspension layer is orders of magnitude smaller than in the saltation layer, which obviously can be compensated for by the possibly orders of magnitude deeper suspension layer at high wind speeds. An important aspect to note here is that the larger particles, which carry most of the mass, remain close to the surface. When the blowing snow layer extends more than 100 m in the atmosphere, the particle size at that elevation is very small and not much mass is transported, even though it could be picked up by remote sensing, or visually appear as a cloud of particles. We now provide more extensive discussion regarding this point, which we think is now also more balanced (see L46-59).

L44: "recent" appears twice.

Fixed, thank you (see L63)!

L80-84: How did you choose those lapse rates, particularly that of ILWR?

2 m air temperature: We aimed to choose a rough and sensible value. -6°C/km is close to many estimates of the atmospheric moist adiabatic lapse rate and is consistent with a bore-hole derived estimate of atmospheric lapse rate (-6.8°C/km) derived from the Antarctic Peninsula (Martin and Peel, 1978).

ILWR: The negative lapse rate is designed to capture the effect of decreasing air temperature with height. Our specific value of -31.25 W/m^2/km is borrowed directly from another Alpine3D based modeling study (Michel et al., 2022).

These references have been added to the revised manuscript (see L107-108 and L112).

L164: Another limitation that you might also consider to discuss is the net domain-integrated erosion-deposition balance equal to zero, due to the absence of transport off the continental margins in this approach, which might be of some significance at the coastal grid points over steep continental margins. I also recommend to elaborate on the expected consequences of 1) assuming that the saltation mass flux accounts for both the contribution of suspension and saltation and 2) not taking into account atmospheric sublimation and its influence on surface sublimation, at least qualitatively, and more particularly for continental-scale simulations for which these processes are believed to be responsible for significant ablation at the surface. We can expect several other processes (vertical advection and sublimation of suspended particles, local turbulence and synoptic wind not related to local topography) to contribute to the net erosion/deposition balance and thus to the spatial variability in

SMB. For instance, could they play a role in the discrepancy between modeled and observed SMB described in Sect. 3.2?

We thank the reviewer for the helpful suggestions to put our approach in a broader context of processes impacting the spatial variability of SMB. We expanded this section considerably to now better discuss this (see L220-227).

Section 2.4: Could you add to the text a recall on if and the, how, snow (microstructural) properties (sphericity, radius, bond radius, density, albedo) are altered by deposition of wind-driven snow?

We agree that the manuscript was missing important information, as also noted by the other reviewer. For this reason, we have added the following text:

"Following erosion and subsequent redeposition, several snow microstructural properties are updated in SNOWPACK according to the "redeposit" scheme presented in Keenan et al., 2021. For example, the density of redeposited snow layers are parameterized according to wind speed (Eq. 4 in Keenan et al., 2021) while sphericity and dendricity are both set to 0.875. The grain radius and bond radius are set to 0.2 mm and 0.05 mm, respectively while albedo is defined by Eq. 7 in Groot Zwaaftink et al. (2013)."

L174-177: One could expect more info on the initialization procedure of SNOWPACK here, not provided in Sect. 2.3. How long is the spin-up before the period studied? Does the initialization procedure also account for the possible influence of erosion and deposition on the firn column? You could also refer to Keenan et al. (2021) and mention the reasonable agreement found with observed firn temperature and density profiles to strengthen your argumentation there.

Thanks for this suggestion, which was also suggested by the other reviewer. We now include more detailed information about the spin-up procedure in this section, including some more detailed references to the Keenan et al. (2021) study (see L240-251).

Section 3.2: To evaluate the benefits of a newly-implemented parameterization, it is usually recommended to compare two simulations, one with and one without. By doing so with two Alpine3D downscalings, you could then quantitatively disentangle the contribution of erosion-deposition from that of SEB to the spatial variability in SMB, and attribute part of the enhanced variability and performance that is due to the modelled erosion-deposition process.

Agreed. The other reviewer likewise raised this concern and we provide the same response here. In the revised manuscript we include an Alpine3D simulation for the year

2015 without horizontal snow redistribution (see newly added Fig. 10). In the revised manuscript, we discuss the effect of simulated snow redistribution on simulated SMB variability, over variability caused by other processes (see newly added Section 3.7, L397-405).

L307: "Simulations results" of what ? please specify.

Upon consideration, we realized that this sentence does not make sense. Thus, we have removed it.

L316: This sentence could be rewritten to present a more correct definition of SMB. You could for instance refer to as "Net snow accumulation" to describe mass gain, as SMB is not necessarily positive.

Good point. We have adopted your suggestion and aim to be more precise with (see L407-408):

"The primary way ice sheets accumulate mass is through net snow accumulation at the ice sheet surface. SMB quantifies the balance between processes which accumulate and ablate mass at the surface of ice sheets."

Conclusion: I think briefly recalling that atmospheric sublimation and suspension are both omitted in your calculation chain would make the conclusion more exhaustive.

Agreed. We now end the Conclusion section as follows (see L433-L436):

"Further, we note that our implementation makes important simplifications by neglecting drifting and blowing snow sublimation as well as horizontal redistribution by way of suspension. In regions where these processes are significant drivers of local SMB, future downscaling efforts would likely be improved by inclusion of these processes into Alpine3D."

**References**

Bintanja, R.: Snowdrift suspension and atmospheric turbulence. Part I : theoretical background and model description, Boundary-Layer Meteorol., 95, 343–368, 2000.

Gromke, C., Horender, S., Walter, B., and Lehning, M.: Snow particle characteristics in the saltation layer, Journal of Glaciology, 60, 431–439, https://doi.org/10.3189/2014JoG13J079, 2014.

Kind, R. J.: Mechanics of aeolian transport of snow and sand, J. Wind Eng. Ind. Aerodyn., 36(2), 855–866, doi: 10.1016/0167-6105(90)90426-D), 1990.

Palm, S. P., Kayetha, V., Yang, Y., and Pauly, R.: Blowing snow sublimation and transport over Antarctica from 11 years of CALIPSO observations, The Cryosphere, 11, 2555–2569, https://doi.org/10.5194/tc-11-2555-2017, 2017.

**A wind-driven snow redistribution module for Alpine3D v3.3.0: Adaptations designed for downscaling ice sheet surface mass balance**

Eric Keenan[1], Nander Wever[1,2], Jan T. M. Lenaerts[1], and Brooke Medley[3]

[1]Department of Atmospheric and Oceanic Sciences, University of Colorado, Boulder, CO, USA
[2]WSL Institute for Snow and Avalanche Research SLF, Davos, Switzerland
[3]Cryospheric Sciences Laboratory, NASA Goddard Space Flight Center, Greenbelt, MD, USA

**Correspondence:** Eric Keenan (eric.keenan@colorado.edu)

**Abstract.** Ice  sheet surface mass balance describes the net snow accumulation at the ice sheet surface. On the Antarctic ice sheet, winds redistribute snow resulting in surface mass balance that is variable in both space and time. Representing wind-driven snow redistribution processes in models is critical for local assessments of surface mass balance, repeat altimetry studies, and interpretation of ice core accumulation records. To this end, we have adapted Alpine3D, an existing distributed snow modeling framework, to downscale Antarctic surface mass balance to horizontal resolutions up to $1\,\mathrm{km}$. In particular, we have introduced a new two-dimensional advection-based wind-driven snow redistribution module that is driven by an offline coupling between WindNinja, a wind downscaling model, and Alpine3D. We then show that large accumulation variability can be at least partially explained by terrain-induced wind speed variations which subsequently redistribute snow around rolling topography. By comparing Alpine3D to airborne-derived snow accumulation measurements within a testing domain over Pine Island Glacier in West Antarctica, we demonstrate that our Alpine3D downscaling approach improves surface mass balance estimates when compared to MERRA-2, a global atmospheric reanalysis which we use as atmospheric forcing. In particular, when compared to MERRA-2, Alpine3D reduces simulated surface mass balance root mean squared error by $23.4\,\mathrm{mm\,w.e.yr^{-1}}$ (13%) and increases variance explained by 24%. Despite these improvements,

our results demonstrate that considerable uncertainty stems from the employed saltation model, confounding simulations of surface mass balance variability.

**1 Introduction**

Ice sheet surface mass balance (SMB) is the difference between mass accumulation and ablation processes at the surface of ice sheets (Lenaerts et al., 2019). Mass accumulation is composed of precipitation as well as condensation and deposition of atmospheric water vapor, whereas ablation processes remove mass from the ice sheet surface via meltwater runoff, both atmospheric and surface sublimation, and evaporation. Additionally, local SMB is influenced by  the process of wind-driven snow redistribution, which we refer to as deposition in the case of  local mass gain and erosion in the case of mass loss (Lenaerts and van den Broeke, 2012).

25 Wind-driven snow redistribution is generally divided into two categories, drifting snow, where snow particles are transported by the wind in the lowermost 2 m of the atmosphere, and blowing snow where snow is redistributed at heights greater than 2 m.

Drifting and blowing snow have been shown to have a substantial effect on Antarctic SMB spatial variability at scales ranging from tens of kilometers (Dattler et al., 2019; Kausch et al., 2020) to meters (Picard et al., 2019). In fact, both Dattler et al.

30 (2019) and Picard et al. (2019) have shown that local deposition can exceed annual precipitation. In addition to redistributing mass, drifting and blowing snow contribute to mass loss by promoting enhanced sublimation as snow particles are entrained in the lower atmosphere (Palm et al., 2017). In spite of drifting and blowing snow sublimation being a source of mass loss, evaluation of model simulation of these processes remains difficult (Amory et al., 2021). The net effect of drifting and blowing snow is preferential deposition in areas of mass convergence at the expense of areas with net divergence (Lehning et al., 2008).

35 Despite our lack of complete physical understanding of the processes which govern preferential deposition at small spatial and temporal scales (Comola et al., 2019b), deposition and erosion can be conceptually summarized as the local convergence of previously eroded snow particles from upwind minus locally eroded snow (Liston et al., 2007). Local erosion is governed by the direct competition between forces that act to erode snow and those which act to anchor snow to the ice sheet surface. Erosive forces, namely surface wind stress, are controlled by atmospheric boundary layer processes

40 (Paterna et al., 2016; Comola et al., 2019a), while cohesive forces are controlled by snow-microstructural properties including grain size and bond strength (Clifton et al., 2006; Melo et al., 2022). This interplay between boundary layer and snow-microstructural processes has historically motivated the development of tightly coupled atmospheric and land surface snow models (e.g., Lawrence et al., 2019; Amory et al., 2021; Sharma et al., 2023).

When the combined effect of surface wind stress and impact force from saltating particles exceeds cohesive forces at the

45 snow surface, saltation of snow particles is initiated or maintained within the lowermost 10 cm of the atmosphere  (Schmidt, 1980; Pomeroy and Gray, 1990). Within the saltation layer,  atmospheric momentum is entrained by snow particles which eventually collide with and subsequently mobilize additional particles at the snow surface. Above the saltation layer, a suspension layer can be present where turbulent eddies generate sufficient upward lift forces that can keep particles afloat against gravity (e.g., Bintanja, 2000). Measurements from Antarctica show that

50 particles in suspension are typically much smaller than those that are transported in saltation (Nishimura and Nemoto, 2005). Correspondingly, mass densities in the suspension layer are found to be orders of magnitude smaller than in the saltation layer. However, with increasing wind speed, the suspension layer can grow orders of magnitude larger than the height of the saltation layer, leading to estimates that the largest part of total mass transport during high wind speed events could be carried in the suspension layer (Pomeroy and Male, 1992; Liston and Sturm, 1998; Mann, 1998). It is difficult to quantify the contribution

55 of suspension to total mass transport on average, with some authors arguing that saltation accounts for 50 – 75% of total wind-driven mass transport (Gromke et al., 2014). However, although deposition and erosion are ubiquitous across much of the Antarctic ice sheet, it is possible that local SMB variability is primarily driven by a small number of high wind speed events which occur only a few times per year.

60    Nevertheless, the interaction between wind transport of snow and the snow surface  is governed by the saltation layer. As saltating particles break apart upon collision with the surface, snow grain fragmentation and rounding are observed, resulting in increased density  (Vionnet et al., 2012; Amory et al., 2021) in a processes referred to as drifting-snow compaction. Reliable model representation of drifting-snow compaction is now particularly attractive owing to recent  advancements in satellite altimetry technology (e.g. CryoSat-2 and ICESat-2). Vertical accuracy, spatial resolution, and ground

65    track repeat frequency have all increased, providing precise measurements of ice sheet surface height change (Smith et al., 2020). However, in order to reliably convert these height changes into mass, particularly over short time scales, we rely on accurate snow and firn density estimates from models (The IMBIE team, 2018). Thus, to confidently assign subtle observed changes in height to quantifiable changes in mass, our models must capture the complex spatial and temporal patterns of deposition and erosion.

70    Current state-of-the-art firn densification models, which are used to convert satellite observed volume changes into mass, successfully capture broad regional variability in firn properties, including density (Ligtenberg et al., 2011; Medley et al., 2022b). However, because of the relatively coarse horizontal resolutions at which they are applied ($5.5 - 35$ km), these models are unable to represent spatial variability in firn processes, including deposition and erosion, at the horizontal scales now sampled by satellites ($< 1$ km). This spatial gap between satellite observations and firn densification models may not be

75    immediately important for mass balance retrievals at continental scales (Verjans et al., 2021). Nevertheless, improved model representation of wind-driven snow redistribution at finer spatial scales can be used to constrain regional to local surface mass balance (Rignot et al., 2011), improve volume-to-mass conversions for repeat altimetry (Shepherd et al., 2012; Zwally et al., 2015), provide the ice coring and radar communities with a mechanism to select representative sampling locations for SMB reconstructions (Kausch et al., 2020), and inform future studies by providing baselines to estimate sublimation of drifting and

80    blowing snow.

To facilitate local SMB estimates and reconstructions as well as repeat altimetry interpretation, we present a new technique for dynamically downscaling Antarctic SMB by building upon the existing Alpine3D v3.3.0 model framework (Section 2.1).  Alpine3D already contains a module for the 3D treatment of drifting and blowing snow (Lehning et al., 2006), but its use requires 3D wind fields at high temporal resolution from an external source, and it is computationally very demanding to

85    run. This motivates the current study to find a simpler, and therefore computationally lighter method to describe wind transport by snow, particularly since the fully 3D approach also does not guarantee good agreement between simulated and observed snow depth in severely wind affected areas (Mott et al., 2008, 2010; Gerber et al., 2017). To this end, we describe the use of WindNinja to downscale wind fields onto local topography (Section 2.2), and demonstrate the use of a one-dimensional snow model to diagnose local erosion (Section 2.3) and then distribute this mass horizontally across adjacent grid cells using a

90    new, two-dimensional advection-based redistribution module (Section 2.4). We then present downscaling results at 1 and 3 km horizontal resolution, and evaluate the added value by quantitatively comparing our results, along with a global SMB product, to a 130 km long airborne radar derived SMB transect in interior West Antarctica over Pine Island Glacier (Sections 3.1 - 3.2, 3.5).

**2  Methods**

 ### 2.1  Alpine3D: Surface mass balance downscaling framework

We use and further develop the existing Alpine3D v3.3.0 model framework (Lehning et al., 2006) to downscale Antarctic SMB processes to a target horizontal resolution of up to 1 km. At its core, the model framework exploits the MeteoIO library (Bavay and Egger, 2014) to handle meteorological preprocessing and downscaling (Section 2.2), SNOWPACK (Bartelt and Lehning, 2002; Lehning et al., 2002b, a), a physics-based land-surface snow model for the detailed description of snow microstructural properties (Section 2.3), and a new module to calculate horizontal mass fluxes between adjacent grid cells (sections 2.4 and 2.6, Fig. 1).

**2.2  Atmospheric forcing: Meteorological downscaling**

At the snow surface, we prescribe hourly MERRA-2 global atmospheric reanalysis (Gelaro et al., 2017) which we have downscaled to the Alpine3D grid using the MeteoIO preprocessing and downscaling library (Bavay and Egger, 2014). These downscaled time series include: 2 m air temperature, relative humidity, incoming shortwave and longwave radiation (ISWR and ILWR), precipitation rate, and 10 m wind speed and direction. 2 m air temperature is first downscaled to the Alpine3D grid using an ordinary kriging algorithm with a lapse rate of -6 $°C\,km^{-1}$ designed to capture a typical atmospheric lapse rate (Martin and Peel, 1978) while relative humidity is spatially interpolated according to Liston and Elder (2006). Precipitation rate and ISWR are then interpolated using inverse distance weighting, with ISWR undergoing a simple correction for slope and topographic shading (Helbig, 2009). ILWR is interpolated by first calculating the hourly average of all MERRA-2 grid cells within the domain, and then applying a constant lapse rate of -31.25 $W\,m^{-2}\,km^{-1}$  (Michel et al., 2022). We follow Keenan et al. (2021), by applying the MERRA-2 mean annual surface temperature as a Dirichlet thermodynamic boundary condition at the bottom of the firn column. This assumption is supported by observations from the dry snow zone of the Greenland ice sheet, where differences between mean annual air temperature and firn temperature at 10m were found to be typically within a few tenths of a degree (Alley and Koci, 1990). Ligtenberg (2014) also shows in a model result that the seasonal cycle in firn temperature disappears around 10m depth. Note that for all topographic calculations, we use an ICESat-2 derived digital elevation model (DEM) (Medley et al., 2022a).

Because reliable simulations of deposition and erosion require accurate wind speed and direction fields (Reynolds et al., 2021), we test two approaches for downscaling wind speed and direction from the relatively coarse (0.5° latitude × 0.625° longitude) MERRA-2 grid. First, we apply the terrain-based index method proposed by Liston et al. (2007) which adjusts wind speed and direction based  on topographic exposure and sheltering. Note that we use the default slope and curvature weighting factors and choose a topographic length scale of 5 km. However, because Reynolds et al. (2021) showed that simulated snow depth better captured observations when forced with WindNinja downscaled wind fields (Forthofer et al., 2014) compared to the relatively simpler terrain-based index methods presented in Liston et al. (2007), we have implemented WindNinja (Version 3.7.1) as an alternative offline wind speed and direction downscaling technique within the Alpine3D modeling framework. WindNinja is a finite element diagnostic model which leverages a mass-conservation solver and DEM

[Figure]

**Figure 1.** Anatomy of an Alpine3D time step: Hourly global atmospheric reanalyis (a) is combined with ICESat-2 derived surface topography (b) to calculate downscaled meteorology (c, d) at each SNOWPACK model grid cell. The SNOWPACK model is then used to calculate snow microstructural properties at each grid cell (d̶e). Finally, the Alpine3D model is used to calculate horizontal mass transport across adjacent grid cells (e̶f) which is then sent back to SNOWPACK for the next time step. Note that in (c,d), downscaled wind speed and 2 m air temperature are shown as an example, while also relative humidity, incoming shortwave and longwave radiation, precipitation rate, and 10 m wind direction are downscaled (see text).

to simulate the mechanical effects of terrain on the flow. In terms of WindNinja model configuration, we have selected the "fine" finite element mesh resolution and chosen "grass" as our surface roughness category (snow is not currently an option). Despite representing a significant increase in complexity compared to other terrain-based interpolation techniques (e.g., Liston et al., 2007), WindNinja is still  computationally cheaper than high-resolution numerical weather models (e.g.

130

the Weather Research and Forecasting (WRF) Model), which solve the non-hydrostatic, fully compressible Navier-Stokes equations at multiple vertical levels (Wagenbrenner et al., 2016), and can therefore be run with reasonable computational resources (Section 2.6).

**2.3 SNOWPACK: One-dimensional physics-based snow model**

135 We use SNOWPACK, a physics-based land-surface snow model, to describe one-dimensional snow and firn processes at each Alpine3D grid cell. SNOWPACK was originally developed for avalanche warning applications, and has been continuously enhanced in order to represent various cryospheric processes including seasonal snow (Sharma et al., 2023), sea ice snow cover (Wever et al., 2020), and polar firn compaction (Groot Zwaaftink et al., 2013; Keenan et al., 2021). In this study, we use the SNOWPACK physics presented in Keenan et al. (2021), with one exception being a new parameterization for surface

140 roughness length, $z_0$ (m), Eq. (1), tuned to observed seasonal variability between winter and summer surface roughness in coastal East Antarctica (Amory et al., 2017). In Eq. (1), $\kappa$ is the von Kármán constant (0.4), $T_{2\,\text{m}}$ is the 2 m air temperature (°C), and $C_{\text{DN10}}$ is the neutral drag coefficient at 10 m, Eq. (2).

$$
z_0 = \begin{cases} \dfrac{10}{\exp \frac{\kappa}{\sqrt{C_{\text{DN10}}}}} & T_{2\text{m}} > -20\,°C \\[2ex] 0.0002 & T_{2\text{m}} \leq -20\,°C \end{cases}
\tag{1}
$$

$$
C_{\text{DN10}} = 2.7 \times 10^{-3} + 9.0 \times 10^{-5} T_{2\text{m}} + 1.5 \times 10^{-6} T_{2\text{m}}^2
\tag{2}
$$

145 In SNOWPACK, snow redistribution is initiated when the friction velocity $u_*$ ($\text{m s}^{-1}$) exceeds the surface threshold friction velocity $u_{*\text{th}}$ ($\text{m s}^{-1}$, Eq. (3)), which is calculated diagnostically as a function of snow microstructural properties including snow grain sphericity $SP$ (0 - 1), radius $r_\text{g}$ (m), bond radius $r_\text{b}$ (m), and coordination number $N_3$ (Lehning and Fierz, 2008).

$$
u_{*\text{th}} = \sqrt{\frac{A\rho_\text{i} g r_\text{g}(SP+1) + B\sigma N_3 \frac{r_\text{b}^2}{r_\text{g}^2}}{\rho_\text{a}}}
\tag{3}
$$

In Eq. (3), $\rho_\text{i}$ is the density of ice (917 $\text{kg m}^{-3}$), $\rho_\text{a}$ is the density of air (1.1 $\text{kg m}^{-3}$), $g$ is the gravitational acceleration

150 (9.8 $\text{m s}^{-2}$), $\sigma$ is a reference shear strength set to 300 $Pa$, while constants $A$ and $B$ are set to 0.02 and 0.0015 respectively. Once drifting snow is initiated, snow layers are eroded from the top of the simulated snow cover until the saltation mass transport rate $Q$ ($\text{kg m}^{-1}\,\text{s}^{-1}$), Eq. (4), is satisfied (Sørensen, 1991). The required local erosion is calculated from $Q$ by scaling to an erosion mass flux $\Phi$ ($\text{kg m}^{-2}\,\text{s}^{-1}$) by dividing $Q$ by a characteristic horizontal length

155 scale $L$, which we call the fetch length, over which the saltating particles in  $Q$ are scaled to calculate local erosion $\Phi$ from the firn layer. $L$ can be conceptually understood to represent the distance over which the originally upwind and now saltating

particles have been eroded from the snow surface (Keenan et al., 2021). These eroded snow layers are then made available to Alpine3D for horizontal redistribution across adjacent grid cells (Section 2.4).

$$\Phi = \frac{Q}{L} = \frac{0.0014\rho_a u_*(u_* - u_{*\text{th}})(u_* + 7.6u_{*\text{th}} + 205)}{L} \qquad (4)$$

160     In contrast to our previous study (Keenan et al., 2021), we revert $L$ from 10 m back to the original SNOWPACK value of 70 m. We make this choice because Alpine3D simulations with a fetch length of 30 m or less were found to significantly overestimate the magnitude of deposition and erosion when compared to observations (Section 3.6). However, because we are not aware of any direct observations, $L$ can be effectively considered a tuning parameter whose magnitude is inversely proportional to the amount of eroded mass accounted for in the saltation mass flux $\Phi$ (Wever et al., 2022). Following erosion
165 and subsequent redeposition, several snow microstructural properties are updated in SNOWPACK according to the "redeposit" scheme presented in (Keenan et al., 2021). For example, the density of redeposited snow layers are parameterized according to wind speed (Eq. 4 in Keenan et al. (2021)) while sphericity and dendricity are both set to 0.875. The grain radius and bond radius are set to 0.2 mm and 0.05 mm, respectively while albedo is defined by Eq. 7 in Groot Zwaaftink et al. (2013).

    Note that we do not explicitly account for blowing snow suspension in our model. We make this pragmatic decision pri-
170 marily  because full prognostic solution of blowing snow transport via suspension requires numerically expensive calculations using wind vector fields at multiple vertical levels  (e.g., Lehning et al., 2006; Sh , while we discussed before that few observational case studies would be available for validation that quantitatively partition between saltation and suspension-driven transport and their eventual effect on deposition and erosion. It is also important to consider that the control provided by the poorly constrained fetch length $L$ ~~can be tuned such that the saltation mass flux
175 $\Phi$ naively accounts for mass fluxes from both saltation and suspension (Keenan et al., 2021)~~would make it possible to locally erode more snow, which naively could be thought of as considering a suspension layer (Keenan et al., 2021), even though the advection distance of snow in suspension is larger given the higher advection speeds and longer airborne times of suspended particles.

    Additionally, it has come to our attention that SNOWPACK's parameterization of $Q$ does not perfectly match the original
180 parameterization proposed by Sørensen (1991). As noted in Vionnet (2012), the parameters 0.0014 and 205 in Eq. 4 reflect units for $Q$ of $\text{g cm}^{-1}\,\text{s}^{-1}$, whereas here we define $Q$ with units of $\text{kg m}^{-1}\,\text{s}^{-1}$. This implementation error in SNOWPACK was introduced by Lehning and Fierz (2008), but since we build upon the previous work by Keenan et al. (2021); Wever et al. (2022) , who showed satisfying results for erosion and deposition calculated using the currently implemented code, we did not correct this error. Practically speaking, this error leads SNOWPACK to underestimate the magnitude of Q compared to the intended
185 parameterization described in Sørensen (1991). However, it is also important to note here that Q is ultimately scaled by the poorly constrained tuning parameter L, such that any changes in the formulation of Q could require a new choice for the fetch length L to maintain similarly good performance. Thus, it is not certain that updating our parameterization of Q would lead to improved or more physically meaningful results. That said, future studies could consider using the updated parameterization

of $Q$ introduced in Sørensen (2004) which Vionnet (2012) showed to produce similar results to our presently dimensionally incorrect parameterization of $Q$ (Vionnet et al., 2014).

**2.4  Numerical treatment of deposition and erosion in Alpine3D**

At each time step, the saltation mass flux $\Phi$ (see Eq. (4)) at each grid cell is scaled to a saltation mass $M_\mathrm{s}$ ($\mathrm{kg\,m^{-2}}$) by multiplying $\Phi$ by the Alpine3D time step $\Delta t_\mathrm{A3D}$ (s) (Eq. (5)).

$$M_\mathrm{s} = \Phi \Delta t_\mathrm{A3D} \tag{5}$$

$M_\mathrm{s}$ is then made available to Alpine3D for downstream redistribution, resulting in a local saltation mass perturbation $\Delta M_\mathrm{s}$ ($\mathrm{kg\,m^{-2}}$) which is positive in the case of net deposition and negative in the case of erosion. We calculate $\Delta M_\mathrm{s}$ by treating wind-driven snow redistribution as a two-dimensional horizontal advection problem (Eq. (6)) where $\boldsymbol{u_\mathrm{s}}$ is the saltation velocity vector ($\mathrm{m\,s^{-1}}$). In our implementation, $\boldsymbol{u_\mathrm{s}}$ represents the saltation particle speed and is defined as parallel to the 10 m wind speed unit vector $\hat{\boldsymbol{U}}_{10\ \mathrm{m}}$ with a magnitude parameterized as a function of $u_{*\mathrm{th}}$ (Eq. (7)) according to Pomeroy and Gray (1990).

$$\frac{\partial M_\mathrm{s}}{\partial t} + \boldsymbol{u_\mathrm{s}} \cdot \nabla M_\mathrm{s} = 0, \tag{6}$$

$$\boldsymbol{u_\mathrm{s}} = 2.8 u_{*\mathrm{th}} \hat{\boldsymbol{U}}_{10\ \mathrm{m}} \tag{7}$$

We solve for $\Delta M_\mathrm{s}$ by numerically integrating Eq. (6) forward in time using a first-order accurate, upwind finite difference scheme with an adaptive sub-time step $\Delta t_\mathrm{CFL}$ (s) in order to ensure numerical stability under the Courant-Friedrichs-Lewy (CFL) condition (Courant et al., 1928). The local saltation mass perturbation at the $i^\mathrm{th}$ sub-time step originating from advection of saltating snow in the $\hat{x}$ and $\hat{y}$ directions $\delta M^i_{\mathrm{s},x}$, $\delta M^i_{\mathrm{s},y}$ ($\mathrm{kg\,m^{-2}}$, Eqs. (8) and (9)), are then calculated with $\Delta X$ being the Alpine3D horizontal grid resolution.

$$\delta M^i_{\mathrm{s},x} = \begin{cases} \frac{\left(M_{\mathrm{s}(x-1,y)} - M_{\mathrm{s}(x,y)}\right)}{\Delta X}(\boldsymbol{u_\mathrm{s}} \cdot \hat{x})\Delta t_\mathrm{CFL} & (\boldsymbol{u_\mathrm{s}} \cdot \hat{x}) > 0 \\ \frac{\left(M_{\mathrm{s}(x,y)} - M_{\mathrm{s}(x+1,y)}\right)}{\Delta X}(\boldsymbol{u_\mathrm{s}} \cdot \hat{x})\Delta t_\mathrm{CFL} & (\boldsymbol{u_\mathrm{s}} \cdot \hat{x}) < 0 \end{cases} \tag{8}$$

$$\delta M^i_{\mathrm{s},y} = \begin{cases} \frac{\left(M_{\mathrm{s}(x,y-1)} - M_{\mathrm{s}(x,y)}\right)}{\Delta X}(\boldsymbol{u_\mathrm{s}} \cdot \hat{y})\Delta t_\mathrm{CFL} & (\boldsymbol{u_\mathrm{s}} \cdot \hat{y}) > 0 \\ \frac{\left(M_{\mathrm{s}(x,y)} - M_{\mathrm{s}(x,y+1)}\right)}{\Delta X}(\boldsymbol{u_\mathrm{s}} \cdot \hat{y})\Delta t_\mathrm{CFL} & (\boldsymbol{u_\mathrm{s}} \cdot \hat{y}) < 0 \end{cases} \tag{9}$$

210 $\Delta t_{\mathrm{CFL}}$ is optimized such that we choose the maximum possible sub-time step while simultaneously satisfying the CFL condition (Eq. (10)).

$$(|\,\boldsymbol{u_s} \cdot \hat{x}\,| + |\,\boldsymbol{u_s} \cdot \hat{y}\,|)\frac{\Delta t_{\mathrm{CFL}}}{\Delta X} < 1 \tag{10}$$

Note that in the case of the final sub-time step, we reduce $\Delta t_{\mathrm{CFL}}$ such that the sum of all $N$ sub-time steps is equal to $\Delta t_{\mathrm{A3D}}$. By summing equations 8 and 9 across all $\Delta t_{\mathrm{CFL}}$ within $\Delta t_{\mathrm{A3D}}$, we are able to finally calculate the local saltation mass 215 perturbation $\Delta M_{\mathrm{s}}$ ($\mathrm{kg\,m^{-2}}$, Eq. (11)) which is ultimately sent to SNOWPACK as a mass flux available at the next time step.

$$\Delta M_{\mathrm{s}} = \sum_{i=1}^{N}(\delta M_{\mathrm{s},x}^{i} + \delta M_{\mathrm{s},y}^{i}) \tag{11}$$

Our upwind finite difference scheme cannot be applied at the edges of a domain, therefore we must prescribe boundary conditions. In this application (Section 2.5) we simulate a finite spatial domain. In order to force our domain to behave more like an infinite domain, we implement periodic boundary conditions (Hames et al., 2022) which implies that deposition and 220 erosion integrate to zero throughout the modeling domain. In areas where a net blowing snow mass influx or outflux with respect to the model domain may occur, for example in coastal regions where mass can be lost to the ocean, as well as in areas with steep topography, this assumption is not expected to hold. However, note that establishing boundary conditions for a blowing snow model in such areas would be very challenging. As discussed before, we may underestimate the effects of snow transport by wind by neglecting the dynamics of the suspension layer, including vertical advection of particles between 225 the saltation and suspension layer, which may impact spatial variability in SMB. Similarly, small scale atmospheric processes, like orographic lifting at scales not represented in the forcing data (Lenaerts et al., 2014), as well as complex flow patterns that can occur in steep terrain (Gerber et al., 2017), may create larger variability in SMB than we can simulate with our approach. Furthermore, we note that our model currently makes a significant simplification by not considering sublimation of snow particles actively entrained in the atmosphere above the snow surface (Amory et al., 2021).

**230 2.5 Pine Island Glacier experimental domain**

As a demonstration of our Alpine3D downscaling framework, we define a 198 km $\times$ 137 km modeling domain centered over the Pine Island Glacier basin in West Antarctica (Fig. 2a, red rectangle). In this domain, surface elevations range from approximately 700–1500 m leading to relatively large surface slopes (mean: $0.64°$, 10% quantile: $0.18°$, 90% quantile: $1.3°$) typical of the Antarctic escarpment zone. Strong and steady winds driven by consistent katabatic forcing, combined with rolling 235 topography, have been shown to drive SMB variability of up to a factor of five (200–1000 $\mathrm{mm\,w.e.yr^{-1}}$) over horizontal scales on the order of 10 km (Dattler et al., 2019). In fact, Dattler et al. (2019) found the largest SMB spatial variability among their 17,500 km of flight line measurements in this domain (Section 2.7). This region therefore provides an opportunity to evaluate our SMB downscaling framework, while still being of modest spatial extent and therefore acceptable computational cost necessary for continuous model development. In terms of model initialization, we borrow from

240   We initialize the model based on the approach employed in our previous study  (Keenan et al., 2021), where good performance for density and temperature was obtained. For each MERRA-2 grid point (Section 2.2) inside the domain, we repeatedly ran a one-dimensional SNOWPACK simulation (Section 2.3)  for the period 1980–2017, while for every iteration taking the simulated firn layer by the end of 2017 as initial conditions

245   at the start of 1980. Those spin-up simulations describe the impact of drifting snow on the surface density and microstructure (Keenan et al., 2021; Wever et al., 2022), but they do not consider spatial snow redistribution. The process was repeated until the firn layer was at least 100 m deep by the end of 2017. We then ran one final simulation from January 1, 1980 until January 1, 2015. Afterward, each Alpine3D model grid point was initialized using the firn properties from the closest MERRA-2 grid  point SNOWPACK simulation. Alpine3D downscaling is then launched at the beginning of the analysis period

250   (2015 in this study), meaning that although we initialize Alpine3D with a spun up firn column, its properties initially reflect the non-downscaled MERRA-2 climate. Note that in order to minimize the relative importance of boundary conditions for drifting snow, upon analysis we remove all grid cells within 15 km of the four domain boundaries, resulting in grid size of 168 km × 107 km (Fig. 2b black dashed rectangle).

**2.6   Computational parallelization and benchmarking**

255   To enable efficient numerical parallelization, the Alpine3D modeling framework supports a hybrid OpenMP/MPI implementation. For benchmarking, we perform calculations on a 1 km horizontal resolution 27,126 km$^2$ domain (Section 2.5) using general purpose compute nodes with 24 parallel CPUs per node. In the case of offline WindNinja wind speed and direction downscaling, we use 1 node (OpenMP, 24 CPUs) for 1 hour per calendar year, resulting in a computational speed of 1,130 km$^2$ yr (CPU hr)$^{-1}$. Whereas for  Alpine3D caclulations, we leverage 4

260   nodes (hybrid OpenMP/MPI, 96 total CPUs) for 18 hours per calendar year, netting  a computational speed of 16 km$^2$ yr (CPU hr)$^{-1}$. WindNinja is therefore at least one order of magnitude  computationally less expensive than Alpine3D, meaning that its added complexity is justified if downscaled SMB estimates are significantly improved.

**2.7   Surface mass balance observations**

265   To evaluate our SMB downscaling, we compare with a 130 km long, radar-derived accumulation transect (Fig. 2b black line T–T') developed by Dattler et al. (2019). This annual-averaged snow accumulation product captures spatial variability in along-track accumulation by tracking fluctuations in radar-derived firn isochrone thickness. Horizontal variations in thickness are then converted to mass using the Herron and Langway (1980) firn density model. At 25 km horizontal resolution, Dattler et al. (2019) set the mean annual (1980–2017) accumulation to that of MERRA-2 atmospheric reanalysis. While at smaller scales, accumu-

270   lation is modified by the combination of firn isochrone thickness and simulated firn density. This product intends to represent the annual-average SMB. However, because of the finite thickness of firn isochrones and spatially variable accumulation rates,

[Figure]

**Figure 2.** Alpine3D modeling and analysis domains in the EPSG 3031 coordinate system: (a) The 198 km × 137 km Alpine3D modeling domain (red rectangle) is centered over the Pine Island Glacier basin in West Antarctica. (b) Surface topography of Alpine3D modeling (red) and analysis (black dashed) domains with the location of the 130 km observed SMB transect (black line, T–T'). Panel (a) was created using the Norwegian Polar Institute's Quantarctica package (Matsuoka et al., 2018).

the annual average is calculated over varying periods depending on the location. No quantitative uncertainty estimates are assigned to the absolute SMB observations, however the authors report an average relative accumulation error of 0.02%.

**3 Results and Discussion**

**3.1 Impact of wind downscaling method on simulated surface mass balance**

To demonstrate the differences between downscaling techniques, we have downscaled MERRA-2 winds from the year 2015 using both the Liston and WindNinja (Section 2.2) algorithms to the 1 km Alpine3D grid (Fig. 3). WindNinja predicts a higher average wind speed than Liston (6.5 m s$^{-1}$ vs. 6.1 m s$^{-1}$) and is more consistent with MERRA-2 (6.6 m s$^{-1}$), which

[Figure]

**Figure 3.** Example of wind speed downscaling for the year 2015: (a) Annual mean MERRA-2, (b) WindNinja, and (c) Liston 10 m wind speed. (d) Annual mean WindNinja minus Liston wind speed. (e) Transect (T–T' in Fig. 2b) of annual mean WindNinja and Liston wind speed as well as transect surface slope.

can be explained by WindNinja's mass conservation solver and Liston's imposed terrain weighting factor of 0.5–1.5 times
280  the original interpolated value (Reynolds et al., 2021) . Notably, both downscaling techniques predict local topography-driven
wind speed variability not captured by MERRA-2, however this variability is in opposite phase, meaning that when WindNinja
predicts relatively high wind speed, Liston predicts locally lower wind speed. Furthermore, WindNinja predicts 2–3 times
larger local topography-driven wind speed variability than Liston, leading us to expect larger SMB variability in Alpine3D
simulations forced with WindNinja winds compared to Liston-driven simulations. Likewise, because predicted wind speed
285  accelerations are out of phase between the WindNinja and Liston downscaling methods, we anticipate the subsequent offline
coupling with Alpine3D to predict opposite patterns of deposition and erosion. Note that our modeling domain (Section 2.5)
lacks the necessary wind speed and direction observations required for a robust evaluation, therefore we focus our evaluation
on downscaled SMB, which ultimately integrates wind speed and direction and can be evaluated against observations (Section
2.7).
290     Owing to the largely contrasting results between the WindNinja and Liston wind downscaling techniques, we have performed
two 2015 Alpine3D simulations, one using WindNinja and one using Liston winds. As discussed above, the WindNinja driven

[revised manuscript text omitted]

Finally, we performed a simulation without activating the newly developed module for wind redistribution of snow (Fig. 10), while keeping the WindNinja downscaling for wind as well as the downscaling approaches for the other variables. We find that
400   in this case, the SMB field (Fig. 10a) is strongly dominated by the large scale gradient in SMB resulting from the MERRA-2

[Figure]

**Figure 9.** Connection between prescribed fetch length and simulated SMB. 2015 simulated SMB with (a) 30, (b) 70, and (c) 110 m fetch length. (d) Transect of observed (red), and simulated SMB with 30 (black), 70 (blue), and 110 m (magenta) fetch length.

forcing data, while lacking the small scale spatial variability found in the simulation with drifting snow shown in Fig. 4b. It illustrates that the Alpine3D simulation does not create any substantial small scale variability in SMB that could arise

[Figure]

**Figure 10.** Comparing the simulation with and without wind redistribution of snow for the year 2015. (a) 2015 Alpine3D SMB without wind redistribution of snow, (b) Transect of observed (red), 2015 Alpine3D with wind redistribution (blue), 2015 Alpine3D without wind redistribution (black) SMB. (c, d) Scatter plot of (c) 2015 with wind redistribution and (d) 2015 without wind redistribution vs. observations.

from other processes. For example, wind speed also controls surface sublimation, but the effect on SMB is not noticeable on larger scales. We conclude that the simulations including the wind redistribution of snow attribute virtually all of the spatial heterogeneity in SMB to snow redistribution processes by the wind (Fig. 4c, d).

**4 Conclusions**

The primary way ice sheets  accumulate mass is through net snow accumulation at the ice sheet surface. SMB quantifies the balance between processes which accumulate and ablate mass at the surface of ice sheets. Throughout the last two decades, regional climate models have been progressively developed to simulate large-scale ice sheet SMB fields.

However, recent advancements in observational capabilities (e.g. repeat satellite altimetry and airborne snow radar) indicate that SMB varies predictably in the presence of rolling topography. Despite the importance of capturing this SMB variability for assigning a mass change to repeat altimetry measurements, regional climate models are not capable of resolving this variability. To begin to close the gap between observations and existing regional climate models, and therefore improve local SMB estimates and support interpretation of repeat altimetry, we have expanded upon the Alpine3D framework in order to dynamically downscale Antarctic SMB.

In our implementation of Alpine3D, we leverage MeteoIO and WindNinja software to downscale meteorology onto the native Alpine3D grid. At each grid cell, the SNOWPACK snow model is used to diagnose snow properties including, density, temperature, grain size, and other snow microstructure parameters. With this information, SNOWPACK calculates a saltation mass flux for each time step and then sends this information to Alpine3D, where horizontal snow redistribution is treated as a two-dimensional advection problem. WindNinja downscaling results suggest that rolling topography with surface slopes between 0.25 - 2.0° are responsible for subtle terrain induced mechanical effects that vary mean wind speeds by $0.1 \mathrm{~m~s}^{-1}$. These wind speed perturbations ultimately drive spatially variable patters of deposition and erosion whose mean magnitude corresponds to 23% of mean annual accumulation.

By comparing 2015 – 2020 mean annual SMB downscaling results over a portion of Pine Island Glacier, West Antarctica to a 130 $\mathrm{km}$ long observational SMB transect, we found that, when compared to MERRA-2, Alpine3D reduces typical SMB errors by 23.4 $\mathrm{mm~w.e.}$ (13%) and increases variance explained by 24%. Furthermore, we show that Alpine3D produces a similarly skillful downscaled SMB product when run at 3 $\mathrm{km}$ horizontal resolution. This suggests that Alpine3D can be run at a coarser resolution than what is primarily shown in this study (1 $\mathrm{km}$ horizontal resolution), therefore considerably reducing computational cost. Finally, despite the promising downscaling results presented in this study, Alpine3D still relies on the fetch length tuning parameter to scale the saltation mass flux. We have shown that the magnitude of deposition and erosion, and consequently SMB, are sensitive to the choice of fetch length. Because it is at the very least difficult, and quite likely impossible, to uniformly optimize the fetch length, we suggest that future studies migrate towards diagnosing the saltation mass flux purely from known quantities including snow properties and surface wind stress. Further, we note that our implementation makes important simplifications by neglecting drifting and blowing snow sublimation as well as horizontal redistribution by way of suspension. In regions where these processes are significant drivers of local SMB, future downscaling efforts would likely be improved by inclusion of these processes into Alpine3D.

*Code and data availability.* MeteoIO, SNOWPACK and Alpine3D are software published under the GNU LGPLv3 license by the WSL Institute for Snow and Avalanche Research SLF, Davos, Switzerland at https://gitlabext.wsl.ch/snow-models. The repository used to develop the versions of MeteoIO, SNOWPACK and Alpine3D used in this study can be accessed at https://github.com/snowpack-model/snowpack with the exact version corresponding to commit f023b9f archived at https://doi.org/10.5281/zenodo.5914787. MERRA-2 atmospheric reanalysis is available at https://gmao.gsfc.nasa.gov/reanalysis/MERRA-2/ and can be retrieved and processed using our workflow available at https://github.com/EricKeenan/download_MERRA2 and archived at https://doi.org/10.5281/zenodo.4560825. WindNinja source code is hosted at

https://github.com/firelab/windninja while the version used in this study is permanently archived at https://doi.org/10.5281/zenodo.4474633. Software used to produce the numerical simulations, analyses, and figures presented in this study are archived at https://doi.org/10.5281/zenodo.5914751 and https://doi.org/10.5281/zenodo.5914727. Airborne snow accumulation radar data used in this study are archived at https://doi.org/10.5281/zenodo.3534315.

*Author contributions.* All authors contributed to experimental design, analysis, and manuscript preparation. Eric Keenan carried out Alpine3D simulations, performed the analysis, and produced the figures. Eric Keenan and Nander Wever both contributed to Alpine3D model development.

*Competing interests.* The authors declare that they have no conflict of interest.

*Acknowledgements.* The authors thank Charles Amory and an anonymous reviewer for their thoughtful comments which helped to improve the manuscript considerably. Eric Keenan, Nander Wever, Jan T. M. Lenaerts, and Brooke Medley acknowledge support from the National Aeronautics and Space Administration (NASA), grant 80NSSC18K0201 (ROSES-2016: studies with ICESat-2 and CryoSat-2). This work utilized the RMACC Summit supercomputer, which is supported by the National Science Foundation (awards ACI-1532235 and ACI-1532236), the University of Colorado Boulder, and Colorado State University. The Summit supercomputer is a joint effort of the University of Colorado Boulder and Colorado State University. Data storage supported by the University of Colorado Boulder "PetaLibrary".